

# Hydrography in the Mediterranean Sea during a cruise with RV Tethys 2 in May 2015

Vincent Taillandier[1], Thibaut Wagener[2], Fabrizio D'Ortenzio[1], Nicolas Mayot[1,+], Hervé Legoff[3], Joséphine Ras[1], Laurent Coppola[1], Orens Pasqueron de Fommervault[1,*], Catherine Schmechtig[4], Emilie Diamond[1], Henry Bittig[1], Dominique Lefevre[2], Edouard Leymarie[1], Antoine Poteau[1], Louis Prieur[1]

[1]Sorbonne Universités, UPMC Université Paris 06, CNRS, LOV, Villefranche-sur-Mer, 06230, France
[2] Aix-Marseille Université, CNRS/INSU, Université de Toulon, IRD, Mediterranean Institute of Oceanography (MIO), UM 110, Marseille, 13288, France
[3] Sorbonne Universités, UPMC Univ Paris 06, CNRS, IRD, MNHN, LOCEAN, Paris, France
[4] Sorbonne Universités, UPMC Univ Paris 06, CNRS, UMS 3455, OSU Ecce-Terra, Paris Cedex 5, France
[+] present affiliation: Bigelow Laboratory for Ocean Sciences, Maine, USA
[*] present affiliation: Laboratorio de Oceanografià Fisicà, CICESE, Ensenada, B.C., Mexico

*Correspondence to*: Vincent Taillandier (taillandier@obs-vlfr.fr)

**Abstract.**

We report on data from an oceanographic cruise, covering western, central and eastern parts of the Mediterranean Sea, on the French research vessel Tethys 2 in May 2015. This cruise was fully dedicated to the maintenance and the metrological verification of a biogeochemical observing system based on a fleet of BGC-Argo floats. During the cruise, a comprehensive dataset of parameters sensed by the autonomous network was collected. The measurements include ocean currents, seawater salinity and temperature, concentration of inorganic nutrients, of dissolved oxygen, and chlorophyll pigments. The analytical protocols and data processing methods are detailed, together with a first assessment of the calibration state for all the sensors deployed during the cruise. Data collected at stations are available under doi:10.17882/51678, data collected along ship track are available under doi:10.17882/51691.

## 1 Introduction

### 1.1 Context of the cruise

The biogeochemical functioning of the Mediterranean Sea is typical of temperate oceanic regions. Seasonal dynamics of phytoplankton follow an increase of biomass in spring even if primary production remains low during the whole year (Marty et al., 2002). The biomass distribution in the Mediterranean Sea is marked by a pronounced east-west gradient (Bosc et al., 2004). This pattern is confirmed by the phenology of the underlying phytoplankton dynamics, that vary from ultra-oligotrophic regimes in the oriental basin to bloom regimes in the north-western basin (D'Ortenzio et al., 2009). An extended study on the geographical distribution of these regimes – related to the Mediterranean bio-provinces – has revealed significant changes at regional scales during the last decades (Mayot et al., 2016). Indeed, the seasonal cycle of biomass concentration turns out to be a reliable indicator of the response of pelagic ecosystems to external perturbations (Siokou-Frangou et al., 2010). Under





increasing anthropic effects and considered as regional hotspot where impacts of climate change will be the largest (Giorgi and Lionello, 2008), the Mediterranean Sea appears to be a key-basin to characterize this indicator under a large panel of possible trophic regimes as well as various physical and chemical environments (Durrieu de Madron et al., 2011).

The seasonal cycles of biomass concentration have mainly been observed from satellite images of ocean colour, thanks to their synoptic coverage of the area. However limited to surface characterization, the link between biomass structuration in the water column and the underlying physical-chemical state over a seasonal scale has only been achieved in few ocean observation sites. The emergence of BGC-Argo floats, which are autonomous profiling platforms embarking biogeochemical sensors and programmed to weekly cycle until 1000-meter depth (Leymarie et al., 2013), now allows to collect oceanographic profiles concomitantly for physical and biogeochemical properties (temperature, salinity, concentration of dissolved oxygen, chlorophyll-a, nitrate). These open new perspectives for the description and comprehension of the biogeochemical functioning of the Mediterranean Sea. For example, the occurrence of phytoplankton blooms can be directly related to the availability of nutrients (D'Ortenzio et al., 2014).

Such technological advances encouraged to set up a dedicated observing system over the Mediterranean Sea with a fleet of a dozen of BGC-Argo floats in operation. This emerging network has been promoted and sustained by French programs such as Equipex-NAOS and the Mermex experiment, as well as at the European level through Euro-Argo infrastructure. However, sensors for biogeochemical properties, even with recent factory calibration, are subject to substantial systematic errors when deployed on BGC-Argo floats, as reported by Bittig et al. (2012) for oxygen measurements or by Pasqueron de Fommervault et al. (2015) for nitrate measurements. In consequence, even if a BGC-Argo float is supposed to be completely autonomous after deployment, reference data for quality assessment of most of its sensors needs to be collected from ship (D'Ortenzio et al., 2014; Johnson et al., 2017). Automatic quality controls are rapidly advancing at the Argo instances (Schmechtig et al., 2015), although most of the methods and protocols are still under assessment. In this context, a dedicated and accurate effort was mandatory to ensure the quality of the data of the Mediterranean observing system composed of BGC-Argo floats.

### 1.2 Objectives and achievements of the cruise

The dataset presented in this paper was collected during an oceanographic cruise carried out in Spring 2015 over the Mediterranean Sea. In our knowledge, it was the first cruise fully dedicated to the maintenance and the metrological verification of an autonomous observing system based on BGC-Argo floats. The objectives of the cruise were twice:

- continue the time series of profile collection that have been set up since 2012 over the Mediterranean Sea, by deploying new BGC-Argo floats and recovering old ones,
- perform the harmonisation of the collection, with a systematic verification of the calibration state for all the biogeochemical sensors that have been or are to be active in the network, and using shipboard measurements as standards of reference.

The choice of a dedicated cruise instead of ships of opportunity was motivated by applying the same protocol of metrological verification for all the floats, using the same instruments and methods of reference. Another crucial point remains on the required flexibility to choose the location of the oceanographic stations, which mainly depended on the state of the network (i.e. the position of the different floats) at the time of the cruise.



The survey covered large parts of the western, central and eastern basins of the Mediterranean Sea with a total route of about 3000nm (Figure 1). The cruise started in Nice (France) on May 12$^{th}$ 2015 and ended up in Nice on June 1$^{st}$ 2015, on board the Tethys 2 which is a 24-meter-long research vessel of the French research national institute (CNRS). The crew was composed of seven mariners and five scientists. The cruise was divided in four legs of about four days: three port calls were programmed

on 18-19 May in Heraklion (Crete, Greece), on 24-25 May in Heraklion, and on 28-29 May in Lipari (Sicily, Italy). The initial cruise planning ended up to be composed of seven oceanographic stations, which represents about two stations of 10 hours per leg. Transects between stations were crossed at 8-11 knots depending on sea conditions.

The characteristics of the vessel appeared to be well fitted to the purposes of the cruise. Its size allowed to handle about twenty floats on deck. It was equipped with frames and winches that allowed to deploy a rubber boat for float recovery, as well as

CTD-Carousel of 12 Niskin bottles. The equipment on board was extensive and up-to-date, with an underway system dedicated to scientific purposes and internet communication facilities needed to reprogram the floats of the network and to get an update of their positions in real time. The wet lab allowed to install the Winkler oxygen titration system and a 9-samples filtration bench for pigments, together with commodities of a freezer, a refrigerator and a tap plumbed to the underway seawater circuit.

Work on board during the transects was dedicated to the surface sampling, together with seawater sample analyses and data

processing. During stations, the CTD-Carousel was deployed and discrete samples were collected for one shallow cast (0-500m) and one deep cast (0-bottom). This sampling strategy has been reduced to a single cast (0-1000m) in case of rough sea conditions, or extended with another cast (0-1000m) for calibration purposes. The number of casts and samples are summarized in Table 1, with a total of 60 pigment samples, 148 oxygen samples, and 154 nutrient samples.

The cruise was prepared in coordination with the Euro-Argo infrastructure so that series of floats were provided by different

European institutes. Overall, three Argo floats from BSH (Germany), three BGC-Argo floats from OGS (Italy), and twelve BGC-Argo floats from LOV (France) were embarked. Hereinafter, only the BGC-Argo floats belonging to LOV are considered. Among these twelve floats, only ten have been deployed, due to technical problems during tests before their deployment.

The standard way consisted in deploying BGC-Argo floats at the end of every stations, as listed in Table 2. Assumed that the

CTD casts and the first BGC-Argo profiles can be considered as co-located in time and space, calibration exercises could have been drawn. Note that the floats were programmed to profile everyday at noon. In consequence, the first deep profile (0-1000m) acquired by the floats could occur the day of the station if deployed early in the morning, or the day after if deployed later, as reported in Table 2.

This protocol of deployment is effective if a working clearance in the area of the station was obtained in order to perform CTD

casts. Unfortunately, this was not the case in the eastern Levantine basin where the definitions of maritime exclusive economic zones are still vague. In consequence, one BGC-Argo float has been deployed without any reference CTD cast in the eastern Levantine (out of the list reported in Table 2). Two other floats were deployed in the same area some days after in the same conditions, however the calibration exercise was performed in the station west Levantine by clamping the floats on the frame of the CTD-Carousel and acquiring a profile (identified as BCN in Table 2) in concomitancy with the reference CTD profile

and discrete samples.



The aims of this paper are to describe the collected dataset. The sensing means and the underlying processing tools for data acquired from the ship and from BGC-Argo floats are detailed in the next section. The description and the way to access to the quality controlled dataset are provided. Finally, a discussion is drawn about the various methodological strategies to update the BGC-Argo network in the Mediterranean Sea and to provide in-situ calibration of the sensors.

## 2 Methods for sensing, processing and quality control

The method employed for measurement (sensor technology, analytical protocol), the method used to process the collected data, and the operated quality control on the final dataset is then presented per parameter (or family of parameters).

### 2.1 Ocean currents

#### 2.1.1 Presentation of the different measurements

Ocean currents were measured with acoustic Doppler current profilers (ADCP), along the ship track and at every station using two dedicated instruments.

The vessel has been equipped since January 2015 with an Ocean Surveyor 75kHz interfaced with a GPS and a gyro-compass. For the cruise, the ship ADCP (hereafter SADCP) was programmed in broadband single-ping profile mode, over 70 bins of 8 m and a blanking distance of 8m. The maximum range obtained was 500m, it was reduced to 250m in the ultra-oligotrophic waters of the Oriental basin.

The CTD-Carousel was equipped with a lowered ADCP (hereafter LADCP) system. It was composed of two RDI's Workhorse Monitors 300kHz, one uplooker was clamped in the upper part of the frame that removed one over 12 Niskin bottles, and one downlooker clamped in the lower frame. The two sensors were synchronized by a command WM15. The system was supplied by an external battery box installed in the lower frame. The LADCP was programmed in narrowband mode with a sampling rate of 1Hz and 20 bins of 8m and a blanking distance null, earth coordinate with tilts 3 beam solution and bin mapping.

#### 2.1.2 Data processing

Data flow from SADCP was archived on board and pre-processed using the manufacturer's software VMDAS, providing 2-minutes averaged velocity profiles. At least once per day, the data collection was uploaded and processed using the software Cascade V6.2 (Le Bot et al., 2011): ocean currents were generated by correcting raw velocity profiles from the ship navigation and attitude. Bottom track was corrected using Gebco 1' bathymetry, however corrections of ocean tides were not applied. Two datasets were composed: one set with a time resolution of 2 minutes for ocean current profiles acquired during stations, one set with a spatial resolution of 1 km for ocean current profiles acquired during transits.

Data flow from LADCP system was processed using the software LDEO IX (Thurnherr, 2014). The architecture of this software allows to replay processing chain with different parameterisations: depth computation whether from bottom track or using the concomitant CTD profile, the threshold of percentage of good values, the assimilation of SADCP data and the weight



of this constraint, whether time resolution (1-second nominal) or vertical resolution (5m bins), adjustment of the variation of magnetic declination.

LADCP data were processed with different levels of complexity. Right after each cast, a first screening of measurements was performed in order to validate the functioning of the system and assess the percentage of good values. When CTD profile

available, a first ocean current profile was computed with refined depth constraint. In a final step, the misfit with a mean SADCP profile during station was attempted to be minimized by iteratively processing LADCP data with this new constraint.

### 2.1.3 Data quality control

An in-situ calibration of SADCP sensor has been achieved during the cruise. An L-shape of 10nm length was crossed back and forth by the ship in calm sea state and moderate speed over a shallow area of the eastern coast of Crete (see Figure 1).

Bottom track was acquired all the time which allowed to compare ocean currents during the way in and the way back, supposedly steady over the 2h duration of the exercise. The two transects were significantly different in amplitude and azimuth. Corrections on misalignment angle (1.1 degree), amplitude factor (1.004) and pitch thresholds (1 and 1.5 degree) for the SADCP have been proposed in order to reduce the misfits between transects. Quality controlled data set of ocean currents along ship track have been post-processed thanks to these corrections.

This post-processed SADCP dataset was also performed during stations in order to assess and improve the quality the 14 LADCP profiles. As reported in Table 3, all the profiles unless at casts 3 and cast 10 are characterized by low velocity errors and acceptable misfits with SADCP profiles. The Table 4 reports the median values of these uncertainties over the 12 acceptable casts using 1-second resolution profiles (approximately 800 ensembles). It is shown that the SADCP constraint does not significantly improves the ocean current estimate in module, but does in azimuth. The quality controlled date set of

ocean currents at stations have been processed with SADCP constraint and binned at 5m resolution.

### 2.2 Seawater temperature and practical salinity

### 2.2.1 Presentation of the different measurements

Temperature and practical salinity properties of seawater were continuously measured at surface along ship track by the underway system of the vessel, and at depth by the underwater unit or by the BGC-Argo floats during the seven stations.

A SeaCat thermosalinograph (SBE21, serial number 3146), hereafter TSG, was mounted in the underway system of the vessel. This instrument is composed of a conductivity cell and a local temperature probe in order to derive practical salinity. A remote temperature probe (SBE38, serial number 0528) interfaced with the TSG was located at the inlet of the underway flow to minimize thermal contamination. A factory calibration of the TSG system was performed within the year preceding the cruise (29 July 2014). The acquisition started on 13 May 0:00 UTC, it was halted during port calls.

The underwater unit was equipped with a CTD (SBE911+, serial number 0329) that interfaced an internal pressure sensor, an external temperature probe (SBE3plus, serial number 2473) and an external conductivity cell (SBE4C, serial number 1313). A factory calibration of the two sensors was performed within the month preceding the cruise (16 April 2015). The GO-SHIP



guidelines (Hood et al., 2010) were followed for the preparation, the maintenance, and the deployment procedure of this instrument package. The two sensors were plumbed in a ducted laminar seawater circuit entrained by a pump. Its intake was located at the lowest part of the instrument package in order to minimize wake effects during descent. The CTD unit was lowered at slow speed (0.5 m.s$^{-1}$ in the first 100 meters of the water column, up to 1 m/s) using the vessel's winch, after a

pause at surface that cleared air from the plumbing. Proper care between casts helped keeping the conductivity sensor clean. The BGC-Argo floats were equipped with factory calibrated CTD modules (SBE41CPs). These modules are designed as for mooring sensors to guarantee long-term stability of temperature, conductivity and pressure measurements. The probes were plumbed in a U-shaped seawater circuit entrained with a pump and taped with anti-foulant devices.

### 2.2.2 Data processing

The TSG data flow of 15 seconds resolution was archived on board together with GPS data flow as unmodifiable hexadecimal encoded files. At least once per day, the data collection was processed to feed a single time series of 5-minutes resolution for UTC time, geolocation, temperature, practical salinity. The standard suite of processing modules of the SBE dedicated software was used: (i) convert raw data to temperature (SBE38) and salinity (SBE21), implementing physical units from sensor frequencies and pre-cruise factory calibrations; (ii) edit outliers using estimates of the standard deviation over blocks of 100-

scans; (iii) window filter time series using median values over 1 minute; (iv) average time series by 5 minute bins.

During stations, seawater properties were sampled at 24Hz with the CTD unit and transmitted on board through an electro-mechanical sea cable and slip-ring equipped winch. At-sea processing of the archive was run after each CTD cast following GO-SHIP guidelines (Hood et al., 2010). Full resolution data were stored as unmodifiable hexadecimal encoded file during acquisition; metadata (serial number and calibration coefficients of the sensors, location and date of the cast, meteorological

conditions) were logged in the file header. The reduction of 24 Hz-signals to 1 dbar-binned vertical profiles was performed with a standard suite of processing modules using the SBE dedicated software: (i) convert raw data to pressure, in-situ temperature, conductivity, implementing physical units from sensor frequencies and pre-cruise factory calibrations; (ii) edit outliers using estimates of the standard deviation over blocks of 100-scans; (iii) low-pass filter (0.15s cut-off) pressure smoothing digitalization noise; (iv) align conductivity and temperature scans to pressure considering sensor response times

and plumbing configuration; (v) compute an adjusted temperature inside the conductivity cell accounting for thermal inertia when heat exchanges are fast (Lueck and Picklo, 1990) ; (vi) edit artificial density inversions when the descent speed is negative due to ship rolls; (vii) derive practical salinity from pressure, conductivity and adjusted temperature following TEOS-10 equation of state (IOC, SCOR and IAPSO, 2010).

Data from BGC-Argo floats were transmitted on land via satellite Iridium communication and disseminated by a dedicated

server. The continuous acquisition at 0.5Hz is performed during the ascent phase of the float, pressure, temperature and practical salinity were decimated then processed before transmission following user's specifications: in the pressure range 0-10dbar the nominal resolution is kept, in the pressure range 10-250dbar, averages by slices of 2dbar were computed; in the pressure range of 250-1000dbar, averages by slices of 10dbar were computed.





### 2.2.3 Data quality control

The pressure measured from the CTD unit was compared on the vessel's deck with a reading of a barometer during port calls. No significant shift was observed that would afford a post-cruise adjustment of this sensor.

There were not any independent samples (such as salinity bottles) or double probes in the CTD unit that would have allowed to assess the stability of the temperature and conductivity sensors. Thus, the quality of CTD data relies on frequent factory calibrations operated on the sensors: a pre-cruise bath was performed in April 2015 (less than one month before the cruise), and a post-cruise bath performed in March 2016 (less than one year after the cruise). The static drift of the temperature sensor between baths was 0.00008°C which is one order to magnitude lower than the theoretical stability of the probe. The static conductivity ratio between baths was 1.0000321 which represents a drift of about 0.0015mS/cm, one order of magnitude lower than the theoretical stability of the probe. Given the reproducibility of the processing method, the uncertainties of measurement provided by the CTD unit should have stayed within the accuracy of the sensors, which is 0.001°C and 0.003mS/cm out of lowered dynamic accuracy cases (such as in sharp temperature gradients).

The data collection of temperature and practical salinity profiles at every station is thus used as reference to assess the two other sensing systems: the TSG and the BGC-Argo floats. Systematic comparisons between the profiles from the CTD unit and the neighbouring data have been lead at every casts.

Considering TSG data set, the median value of temperature and practical salinity over a time window of 1h around profile date was extracted from the 5-minute resolution time series. The comparison with the surface value from profile is reported in Table 5. It is shown a spread distribution of misfits for temperature, with an average 0.009°C, and a narrower distribution of misfits for practical salinity with an average of 0.007. Given the nominal accuracy expected by the TSG system and in absence of systematic marked shift in the comparison, no post-cruise adjustment has been performed. The uncertainty of measurement in the TSG data set should have stayed under the 0.01°C in temperature and 0.01 in practical salinity.

Considering BGC-Argo floats, the comparison with CTD profiles was performed over the layer 750dbar – 1000dbar, in a layer less influenced by non-concomitancy between profiles. The misfits between temperature measurements and practical salinity measurement at geopotential horizons were computed and median values provided for every BGC-Argo floats. The median offsets are reported in Table 2. They amplitudes do not reach the threshold of 0.01°C in temperature or 0.01 in practical salinity, unless in two cases. A large temperature offset stand for lovbio089d, which might be due to long duration between profiles. A large practical salinity offset was reported for lovbio083d however deployed in exact concomitancy with the CTD profile.

### 2.3 Oxygen concentration

### 2.3.1 Presentation of the different measurements

Concentration of dissolved dioxygen ($O_2$) in seawater, hereafter described as oxygen, was measured with three techniques: the classical iodometric Winkler method, an electrochemical oxygen sensor, optical oxygen sensors.





Oxygen concentration was measured following the Winkler method (Winkler, 1888) with potentiometric endpoint detection (Oudot et al., 1988) on discrete samples collected with Niskin bottles. For sampling, reagents preparation and analysis, the recommendations from Langdon (2010) have been carefully followed. Seawater samples were collected as soon as possible after each cast, inside calibrated PyrexTM flasks (ca. 100mL) with ground glass stoppers. One millilitre of the pickling reagents

was immediately added and, after shaking, the samples were stored in dark before titration. The titration was achieved within a minimum of 4 hours and a maximum of 24 hours, by batches of one or two casts (11 to 22 samples). The titration was performed within the calibrated flasks using a Metrohm 888 Titrando titrator with an exchangeable burette of 10mL, a platinum combined electrode, and a Metrohm 801 stirrer. The pickling reagents, the sulphuric acid and the iodate solution were dispensed with calibrated pipettes or dispensers. The $KIO_3$ standard solution was prepared with dried potassium iodate

dissolved in ultrapure water. The exact concentration ($0.01648 \pm 0.00001$ mol.$L^{-1}$) of the homemade $KIO_3$ standard was determined by titrating it against a potassium iodate standard solution of 0.0100N (WAKO).

Oxygen concentrations have been measured using electrochemical sensors (Kanwisher, 1959). The sensing technique is based on a Clark cell where the amperometric reduction of oxygen that flows through a gas-permissive membrane in contact with seawater is measured. The cathodic limit current is proportional to the diffusive flux of oxygen to the electrode and is thus

related to the oxygen concentration in seawater. A Seabird SBE43 (serial number 0587) electrochemical sensor was interfaced with the CTD unit. This sensor was plumbed in the pumped circuit, at the outlet of the conductivity cell, in order to ensure a sufficient flow of seawater on the membrane to avoid dynamic errors. The output signal is a voltage proportional to the temperature compensated cathodic current measured by the cell. A factory calibration of the sensor was performed within the month preceding the cruise (14 March 2015), providing 13 coefficients to derive oxygen concentration from the output voltage.

Oxygen optical measurements (also called optode measurements) have been developed for applications in natural waters since two decades (Klimant et al., 1995). Commercially available sensors for profiling applications (with reduced time responses) are available since more than ten years (Kortzinger et al., 2005). Oxygen optical measurements are based on the measurement of the luminescence quenching by oxygen of an immobilized luminophore on a sensing foil (a platinum porphyrin complex into a silicone matrix). By quenching the luminescence, oxygen reduces the lifetime of the luminescence after excitation by a

short pulse of light. In optodes, the luminophore is excited by an intensity modulated blue light beam from a LED. The quenching of luminescence will induce a phase shift on the luminescent response of the foil. To ensure stable measurements, a reference phase reading is made by the use of a red LED that do not produce fluorescence in the foil. The phase shift induced by the oxygen quenching is related to the amount of dissolved oxygen that diffuses in the foil. When the foil is in contact with seawater, the optode provides measurements of oxygen concentrations in seawater. Two types of optodes were used during

the cruise:

•         one Rinko III dissolved oxygen sensor from JFE Advanced Co. Japan (serial number 171) was interfaced with the CTD unit using the analog output voltage,

•         Aandeeraa 4330 optodes were mounted on every BGC-Argo floats.



### 2.3.2 Data processing

For Winkler titration, thiosulfate concentration and the reagent blank were assessed at least in triplicate at the begin of each batch of measurements, then every 11 samples and at the end of the batch (see Table 6). The reagent blank determination and the standardization of the thiosulfate solution were performed at the same time by titrating two successive additions of 1 mL

of $KIO_3$ standard in distilled water. The reagent blank was determined as the difference between the first and the second thiosulfate added volume for titration. The second titration was used to standardize the thiosulfate. The precision of the Winkler measurements was estimated by reproducibility tests based on 5 or 6 replicates (see Table 7). The standard deviation on the replicate measurements was lower than 0.4 $\mu mol.kg^{-1}$. The titration volumes were converted to oxygen concentrations in $\mu mol.kg^{-1}$ by following the calculation procedure proposed in Langdon (2010). Each batch of Winkler measurements was

calculated with the average blank volume and thiosulfate concentration measured over the entire batch. The temperature of sample at pickling has not been systemically recorded and the potential temperature derived from CTD measurements has been used instead, in order to estimate the seawater density at the closing of the vials.

The sensor signal of the SBE43 was aligned to temperature and pressure scans considering a unique plumbing configuration for cruise, by an advance of 3s. The raw signal was then converted to an oxygen concentration with 13 calibration coefficients.

The method is based on the Owens and Millard (1985) algorithm that has been slightly adapted by Seabird in the data treatment software using a hysteresis correction. A new set of calibration coefficients for this sensor has been determined after the cruise, it has been used to post-process the whole dataset. Only three coefficients (the oxygen signal slope, the voltage at zero oxygen signal, the pressure correction factor) among the 13 determined by the pre-cruise factory calibration of the sensor were adjusted with the following procedure. The oxygen concentrations measured by Winkler were matched with the signal measured by the

sensor at the closing of the Niskin bottles. The three values were fitted by minimizing the sum of the square of the difference between Winkler oxygen and oxygen derived from sensor signal. Outliers were discarded when the residuals exceeded 2.8 standard deviation of the residuals until no more outliers remain.

The Rinko optode provided continuous voltage output at 24Hz, which has been directly converted to an oxygen concentration with the Matlab code developed by the manufacturer. The original calibration coefficients have been used. To process the

results, the temperature measured from the CTD unit was preferred to the built-in temperature of the sensor.

The Aanderaa optodes 4330 output signal is a C1 raw phase (phase from the blue light excitation), a C2 raw phase (phase from the red light excitation), and the optode temperature. The calculation of oxygen concentrations from the optode signal follows the recommendations of Thierry et al. (2016). The calibrated phase estimated from the C1 and C2 raw phases is converted in oxygen concentration by the Stern-Volmer equation proposed by Uchida et al. (2008) using seven calibration coefficients (the

so-called Stern-Volmer-Uschida coefficients). The oxygen concentration is then corrected from salinity and pressure effects. The pressure compensation is estimated following Bittig et al. (2015) with a step of phase adjustment. Finally, concentrations are expressed in $\mu mol.kg^{-1}$ by using the potential density derived from the CTD measurements of BGC-Argo floats.



### 2.3.3 Data quality Control

Winkler measurements on discrete samples collected during upcasts were considered as the reference oxygen value because winkler measurements rely inderectly on a certified reference material ($KIO_3$ standard) and that precision on replicate measurements is lower than 0.4 µmol.kg$^{-1}$. The reference Winkler measurements were used to adjust the calibration

coefficients of the CTD oxygen sensor (SBE43), as described behalf. The corrected oxygen profiles during downcasts from the SBE43 at stations were considered as the reference profile for optode measurements from BGC-Argo floats. This quality control was based on the downcasts profiles at 1dbar resolution collected whether by the electrochemical sensor SBE43 or the optode RINKO.

Residuals with Winkler measurements were expressed as the difference on an isobaric horizon between the sensor oxygen and

the Winkler oxygen. A sensor error was estimated as the root mean square error on the residuals. Results are reported in Figure 2, where the residuals over the entire cruise are plotted as a function of time and depth. Residuals appear higher and more variable in the upper part of the water column, most probably due to enhanced oxygen gradients and changes on isobaric horizons between downcasts and upcasts. For electrochemical measurements, no significant offset or drift were observed; the sensor error over the entire cruise is 2.4 µmol.kg$^{-1}$. For optode measurements, the sensor error over the entire cruise was 6.0

µmol.kg$^{-1}$ and a systematic offset of 4.8 µmol.kg$^{-1}$ was observed. Moreover, a significant increase of the residuals with depth (0.0022 µmol.kg$^{-1}$.dbar$^{-1}$) was observed below 200 dbar.

It has been reported that a systematic shift in the optode calibration coefficients can occur during storage and shipment of the sensors (Bittig et al., 2012). In order to compensate this potential shift, float oxygen measurements were corrected based on a reference profile as in Takeshita et al. (2013). A slope and offset value were determined for every deployed optodes in order

to adjust a posteriori the calculated oxygen values from the raw signals. The adjustment of optode values were performed using a linear model, below the first 50 dbar to avoid strong variability in the surface layer, and behalf the last 50 dbar to get rid of possible hooks at the bottom of profiles. The results, reported on Table 2, show a consistent correlation between the two sensors with slopes close to 1 and offsets ranging from -14 µmol.kg$^{-1}$ to 11 µmol.kg$^{-1}$.

### 2.4 Chlorophyll-a concentration

**2.4.1 Presentation of the different measurements**

The chlorophyll-a concentration ([Chl-a], sum of chlorophyll-a, divynil chlorophyll-a and chlorophyllide-a) in seawater was measured with two methods: the high performance liquid chromatography (HPLC), and the fluorescence.

The HPLC method is used to estimate the [Chl-a] in discrete seawater samples collected from the TSG system or withdrawn from Niskin bottles. For this, 2.27 L of the seawater samples were filtered onto glass fibre filters (GF/F Whatman 25 mm), and

all filters were stored in liquid nitrogen then at -80°C to further analysis in laboratory. The chlorophyll-a and other accessory phytoplankton pigments were then extracted from the filters in 100% methanol, disrupted by sonification and clarified by filtration (GF/F Whatman 0.7µm) after 2 hours. Extracts were injected (within 24 hours after beginning of the extraction) on



a reversed phase C8 column and 24 pigments were separated, identified and quantified according to the HPLC analytical protocol described by Ras et al. (2008).

Fluorometers provide continuous detection of chlorophyll-a. Three kinds of sensors were used during the cruise: a Chelsea Aqua Tracka III fluorometer (serial number 088193) interfaced with CTD unit, ECO WetLabs fluorometers that equipped
every BGC-Argo floats, and a Turner fluorometer (serial number SN6241) plumbed in the TSG system of the vessel. The sensing mean is based on the fluorescence concept: irradiated by blue light, the chlorophyll-a absorbs and re-emits in the red part, and the re-emitted signal (i.e. the fluorescence) is considered proportional to the [Chl-a] (Lorenzen, 1966). However, to retrieve the exact [Chl-a] through the raw fluorescence signal, a calibration of the signal is necessary.

Note that fluorescence is affected by non-photochemical quenching, the mechanism employed by phytoplankton to protect
from effects of high light intensity. As a result, amplitude of signal is reduced for an identical [Chl-a] when the measurement is performed under sun light exposure in the sea surface layer.

### 2.4.2 Data processing

The [Chl-a] is derived from raw fluorescence signal by a linear model using two calibration coefficients: an offset that corresponds to the value of the signal in the absence of [Chl-a], a scaling factor to align the signal on the exact in situ [Chl-a].
These calibration coefficients are generally provided by the manufacturer, but an adjustment using in-situ measurements of [Chl-a] is recommended. The calibration method was based on the alignment of the fluorescence signal to exact in-situ discrete measurements of [Chl-a] provided by the HPLC method. For this, a least square linear regression was used with simultaneous measurements of [Chl-a] from fluorescence at the time, location and depth of collected seawater samples analysed by HPLC. The statistics associated to the linear regression were used as a quality control of the calibration.

Fluorometer derived [Chl-a] profiles at CTD casts were processed as follows. As a pre-processing step, the raw fluorescence measurements were corrected from possible non-photochemical quenching following the procedure of Xing et al. (2012). The linear regression was done with 61 simultaneous measurements of [Chl-a] determined by HPLC and the fluorometer (Figure 3). In order to assure a correct linear regression, outliers were evaluated by using the Cook's (1977) distance statistics (three outliers detected, red points in Figure 3). The resulting coefficients were an offset of 0.168 mg.m$^{-3}$ and a slope of 4.016. An
alternative estimation of the offset has been performed by computing the median value of raw fluorescence profiles in the last 50m of every profile. Indeed, when the water column is stratified (it was always the case here), the availability of light is not enough to allow the presence of active phytoplankton cells, thus the fluorescence signal should be null. This estimation considering all the fluorescence profiles provides an offset of 0.160 ± 0.004 mg.m$^{-3}$.

As for CTD casts, the raw fluorescence measurements from BGC-Argo floats were corrected from possible non-photochemical
quenching, and offsets were determined as median values of raw fluorescence in the last 50m of the profiles. The estimated offset values are reported in Table 2. Once offsets were adjusted, the linear regressions were performed with seven or eight simultaneous measurements of [Chl-a] obtained by HPLC at the float deployment. The estimated slopes are reported in Table





2. In average from all the calibration conducted, slopes range from 0.49 to 0.67 with an average value of 0.58; offsets range from -0.02 to 0.04 mg.m$^{-3}$ with an average value of 0.02 mg.m$^{-3}$.

Considering fluorometer derived [Chl-a] along ship-track, a post-cruise estimation of the calibration coefficients for the Turner Fluorometer has been achieved. The linear regression was done with discrete seawater samples expressively collected at night
(between 19 pm to 5 am) to avoid the non-photochemical quenching. As reported in Figure 3, the obtained calibration coefficients were an offset of 0.059 mg.m$^{-3}$ and a slope of 4.831. The raw fluorescence measurements were included in the TSG data flow of 15 seconds resolution. Its processing followed the same steps as for ship-track temperature and salinity: (i) convert raw data to fluorescence, implementing physical units using the post-cruise calibration coefficients; (ii) edit outliers using estimates of the standard deviation over blocks of 100-scans; (iii) window filter time series using median values over 1
minute; (iv) average time series by 5 minute bins.

### 2.4.3 Data quality control

In the Table 8, the list of quantified pigments and their limits of detections (calculated in ng per injection and as the concentrations corresponding to a signal-noise ratio of 3) are provided. Different quality control steps were applied during HPLC analysis, data processing and on the final dataset. During HPLC analysis, parameters such as the stability of the baseline,
the injection precision and the pressure were monitored regularly in order to detect potential anomalies in the analytical process. During data processing, chromatographic parameters were checked, including critical pair resolution, baseline noise, and peak width or retention time precision. Spectral data for the different peaks were verified and used for identification purposes and peak purity assessment. The final pigment database underwent a visual verification step for each pigment of every vertical profiles and quality flags were assigned for each value. When a pigment was not detected, the quality flag "5: not reported"
was used. The visual check confirms that the identification and quantification of all the samples did not present any issues, such as coelution problems or baseline noise thus leading to potential uncertainties.

Considering fluorescence measurements collected on CTD casts, the high coefficient of determination ($r^2$=0.96) for the linear model denotes a very good regression with HPLC data (see Figure 3, upper panel). The pair of calibration coefficients were applied in the post-processing of fluorescence data at every casts.
Considering fluorescence measurements collected by BGC-Argo floats, a good alignment with in-situ data was reached with coefficients of determination higher than 0.75 (Table 2). Moreover, the homogeneity of slopes among the series of new sensors (thus recently factory calibrated) gives an insight on the gain (between 1.8 and 2) to be applied afterwards on fluorescence data (Roesler et al., 2017).

Considering fluorescence measurements collected on the TSG system, its range along the ship track appears very narrow (from
0.035 to 0.112 mg.m$^{-3}$). In addition, a low number of simultaneous HPLC measurements is available (only 9 samples), and the coefficient of determination of the linear is lower than 0.70 (Figure 3, lower panel). Thus, the calibration effort performed is certainly not enough to be fully confident in the adjusted coefficients, although they have been applied to the TSG time series.



### 2.5 Nitrate concentration (and other nutrients)

### 2.5.1 Presentation of the different measurements

Concentration of nitrate (NO3-) ions in seawater were measured with two techniques: the classical colorimetric method in conjunction with nitrite, phosphate and silicate concentrations, and an optical nitrate sensor.

Nutrient samples were collected a few minutes after each cast directly at the tap of the Niskin bottles without tubing. Samples were transferred to 20mL polyethylene flasks (scintillation flasks). They were poisoned with 50μL of a HgCl2 solution to reach a final concentration 20μg.L$^{-1}$ as recommended by Kirkwood (1992). The samples were stored in a refrigerator at 5°C until analysis. Nutrient concentrations remain stable for several months when poisoned samples are stored in the dark at 5°C (Kirkwood, 1992). All nutrient samples were analysed by a standard automated colorimetric system on a Seal Analytical

continuous flow AutoAnalyser III (AA3) at the Observatoire Oceanologique de Villefranche-sur-Mer. The automated system was set up following Aminot and Kerouel (2007). Nitrite (NO$_2^-$) ions were analysed via the formation of a coloured Azo dye (Bendschneider and Robinson, 1952). Nitrate ions were analysed as nitrite after reduction on a copper cotted cadmium column following Morris and Riley (1963). The nitrate concentration was estimated from nitrate with nitrite measurements, from which the nitrite concentration was subtracted. The detection limit for nitrate and nitrite was 0.01μM. Phosphate (PO$_4^{3-}$) ions were

analysed via the formation of molybdenum blue (Murphy and Riley, 1962) with a detection limit of 0.02μmol.L$^{-1}$. Silicate (Si(OH)4) ions were analysed via the formation of yellow silicomolybdic acid (Strickland and Parsons, 1972) with a detection limit of 0.02μmol L$^{-1}$.

Optical sensor measurements were performed on BGC-Argo floats. Historically, several methods have been developed, over the past 50 years, for the direct determination of nitrate concentrations in natural waters, without chemical manipulations (e.g.

Bastin et al., 1957). The measurement principle is based on the absorption of light at ultraviolet wavelengths and the deconvolution of nitrate concentrations from the observed spectrum (Thomas et al., 1990). In seawaters, the ultraviolet spectrum is composed of three main components: bromide, nitrate and to a lesser extent organics particles (Ogura and Hanya,1966). Sensors using miniaturized ultraviolet spectrophotometers now allow continuous measurements of absorbance spectra and estimations of nitrate concentrations (Johnson and Coletti, 2002). The BGC-Argo floats deployed during this cruise

were equipped with SUNA-V2 (Submersible Ultraviolet Nitrate Analyser) sensors commercialized by Satlantics.

### 2.5.2 Data processing

Nitrate concentrations are derived from absorbance spectra using the the TCSS algorithm (Temperature Compensated Salinity Subtracted) set up by Sakamoto et al. (2009). It was demonstrated that the method could be used to resolve annual cycle in mesotrophic and high nutrients environments (e.g. D'Asaro et al., 2008), and to measure nitrate variability driven by mesoscale

events (Johnson et al., 2010). In the Mediterranean Sea, because of specific conditions of low nitrate concentrations and high salinity (thus high bromide concentrations), optical measurements of nitrate were extremely delicate (D'Ortenzio et al., 2014). Recently, a specific algorithm adapted from Sakamoto et al. (2009) substantially improved the estimation of nitrate




concentration in the Mediterranean Sea (Pasqueron de Fommervault et al., 2015). Major changes in the algorithm involve application of a pressure-dependent correction to the bromide spectrum (2% per 1000dbar), interpolation of temperature and salinity values at the position of the sensor (located about 1.5 meter below the CTD sensor), and the modification of the wavelength offset (see Sakamoto et al., 2009 for details) which is now fixed at 208.5nm.

The BGC-Argo floats deployed during the cruise were transmitting the raw data of the SUNA (i.e. absorbance spectrum from 217 to 250 nm), which allowed a post-processing with the algorithm of Pasqueron de Fommervault et al. (2015).

### 2.5.3 Data quality control

A spike test was applied following the procedure described in Pasqueron de Fommervault et al. (2015). A test for saturation was also performed based on the raw absorption spectrum. Nitrate concentration data computed from a spectrum for which

more than 25% of the channels saturate (i.e. reached the maximum value of 65535 numerical counts) were discarded. This was the case of one BGC-Argo float (lovbio093d).

The SUNA sensors also undergo offset and gain (Johnson et al., 2013) that were corrected using as reference the measurements on discrete samples. Given that surface nitrate concentrations in May and June in the Mediterranean Sea stand below the limit of detection of the sensor (Pasqueron de Fommervault et al., 2015), an offset was computed as the difference between an

assumed surface concentration of zero and the mean nitrate value measured from 5 to 30m. A gain was then calculated with a match up between sensors measurements and nitrate concentration at discrete depths. Gain correction was applied only if the misfits between sensor derived and reference concentrations below 950dbar did not exceed 10% of the deep reference value. The correction coefficients per BGC-Argo float are reported in Table 2. A slope of 1 was estimated for most of the cases, and the offsets ranged from -2.70 μmol L$^{-1}$ to 3.90 μmol L$^{-1}$.

### 20  3 Data availability

The final data set concatenates the different collections during the cruise, which are vertical profiles and bottle samples at CTD casts, along track measurements at surface and at depth. This data set benefits for post-cruise corrections described in the previous sections. A unique convention has been used to identify bad data, absent data, or not reported data: they have been assigned to the value -999.

The quality control provided to discrete samples collection has been assigned with a quality flag. The quality code set up for WHP bottle parameters data has been used, in particular: "2: Acceptable measurement", "5: Not reported", "Sample not drawn for this measurement from this bottle".

Data are published by SEANOE operated by SISMER within the framework of the in the information system ODATIS. Data at stations are available under doi:10.17882/51678, data along ship track are available under doi:10.17882/51691.



## 4 Conclusions

The dataset presented in this paper has been collected in the framework of an emerging in-situ observing system in the Mediterranean Sea. In order to characterize the seasonal cycles of phytoplankton dynamics and the biogeochemical functioning of the Mediterranean Sea, this network of twelve BGC-Argo floats collects physical and biogeochemical properties

(temperature, salinity, concentration of dissolved oxygen, chlorophyll-a, nitrate) along 1000-meter depth profiles with a weekly sampling rate. In spring 2015, shipboard measurements have been acquired with the objective to provide a reference dataset for each core parameter of the in-situ observing system. This dataset offered the possibility to perform metrological verification of the deployed sensors, considering the misfits between the first profile of the float and the shipboard data.

First of all, the presented dataset provides an in-situ characterization of the environmental conditions in which the exercises of

verification have been conducted. Thanks to ocean currents and surface hydrography collected along the ship track, a first assessment of the circulation patterns neighbouring every stations can be drawn. Complemented with satellite observations (altimetry, images of sea surface temperature or ocean colour), the degree of stability of the water column would be diagnosed in order to rely (or not) on the co-location in space and time of the BGC-Argo float profile with reference data.

Second, the presented dataset provides material for a systematic calibration of the biogeochemical sensors active in the

network. It has been shown (Table 2) the crucial role of this operation on newly deployed sensors. Concerning the oxygen optode sensors, their linear response does not seem to be affected, however offsets reaching amplitudes of 15 $\mu$mol.kg$^{-1}$ have been reported, without any systematic bias among the set of sensors. Concerning fluorometer sensors, offsets can be corrected considering dark values at depth, however the amplitudes of the signals appeared to be overestimated by a factor between 1.5 and 2 depending on the sensor. Concerning nitrate sensors, their behaviour at deployment is similar to the optodes in terms of

calibration, with a sensor-dependent offset until 4 $\mu$mol.L$^{-1}$ of amplitude. Overall, the biogeochemical sensors embarked on the BGC-Argo floats have revealed inherent calibration shifts at deployment. This is in agreement with recent works on fluorometers ECO WetLabs (Roesler et al., 2017) and on oxygen optodes (Bittig et al. 2015, Bittig and Körtzinger, 2015).

The presented dataset is relevant to provide a robust evaluation of the calibration state of biogeochemical sensors, at the beginning of their mission. In addition, if an equivalent dataset is collected at the end of the mission when the BGC-Argo floats

are recovered, reference data at recovery, the sensor drifts would be properly assessed from pre-mission and post-mission calibration states. This objective appears prerogative to enable in fine the harmonisation between all the time series observed by the network.

This dataset offers a first attempt to evaluate the uncertainties that come up in the verification exercises. When measuring misfits between shipboard measurements and the first profile of the BGC-Argo floats, the natural variability of the environment

can affect their complete attribution to calibration shifts. The expected variations would depend on the type of parameter, on the depth of inter-comparison, on the duration or on distance between profiles. Among the BGC-Argo floats deployed during the cruise, two benefited for a verification exercise in perfect concomitancy as they were clamped on the CTD-Carousel. The first results show reduced dispersion in function of depth for all the parameters. This dispersion criterion needs to be assessed



more carefully with different types of match-up, in function of local environmental conditions and duration or distance with the first profile.

Preliminary conclusions stress the importance of evaluating the calibration state of the biogeochemical sensors, as well as their possible drift during several years of mission. The dataset collected during the cruise of May 2015 provided the relevant material to perform such exercises of metrological verification, and motivates its extension for the future deployments. The cruise also reveals unintentionally the possibility to perform a pre-deployment verification exercise some days before the beginning of its mission. The floats with newly verified sensors have been deployed close to recovered ones in order to continue their time series and to retrieve post-mission calibration states. If the propagation of reference between missions is satisfactory, such a protocol could be applied on conventional oceanographic cruises as it demands one station of metrological verification with floats mounted on the CTD-Carousel and changes of route for float deployment and recovery operations.

**Author contribution**

This dataset was collected by VT, TW, FDO, HLG and NM. TW analysed the oxygen samples, JR analysed the pigment samples, ED analysed the nutrient samples. Data processing and quality control were achieved by HLG for ocean currents and TSG, by VT for seawater hydrological properties, FDO and NM for chlorophyll-a concentration, TW, LC, HB and DL for oxygen concentration, OPF and FDO for nitrate concentration. VT, AP and EL set up BGC-Argo floats deployments and recoveries. Data management and availability was achieved by CS. VT and TW prepared the manuscript with contributions from FDO, NM, JR, LP and OPF.

**Competing interests**

The authors declare that they have no conflict of interest.

**Acknowledgments**

We would like to thank Captain Dany Deneuve and the crew of RV Tethys 2. These observational efforts were supported the project Equipex-NAOS, the Euro-Argo infrastructure, the program MerMex, and the project BAMA funded by LEFE/GMMC. We gratefully acknowledge their support.

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



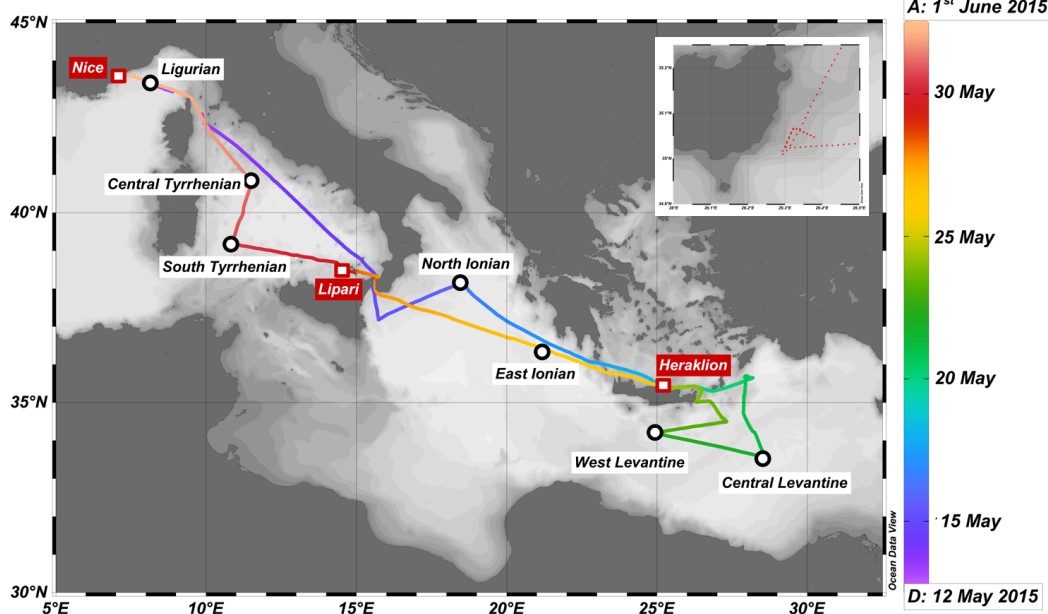

**Figure 1: Cruise track plotted on a time line (colorbar). Port calls marked by red squares, stations are marked by black circles. Zoom of the L-shape track in the eastern coast of Crete.**





| STATION | CAST | DATE UTC | LATITUDE | LONGITUDE | PROFILE DEPTH (m) | BOTTOM DEPTH (m) | # SAMPLE PIGMENTS | #SAMPLE OXYGEN | # SAMPLE NUTRIENTS |
|---|---|---|---|---|---|---|---|---|---|
| LIGURIAN | 1 | 12/5/15 20:10 | 43° 33.52' N | 7° 27.78' E | 1662 | 1684 | 5 | 11 | 11 |
| NORTH IONIAN | 2 | 16/05/15 03:41 | 38° 10.44' N | 18° 30.12' E | 500 | 3038 | 8 | 5 | 11 |
|  | 3 | 16/05/15 05:35 | 38° 10.96' N | 18° 30.16' E | 2990 | 3038 | 0 | 11 | 11 |
| CENTRAL LEVANTINE | 4 | 21/05/15 12:21 | 33° 33.90' N | 28° 27.99' E | 500 | 2959* | 8 | 11 | 11 |
|  | 5 | 21/05/15 14:14 | 33° 33.76' N | 28° 28.50' E | 1240 | 2959* | 0 | 11 | 11 |
| WEST LEVANTINE | 6 | 22/05/15 10:33 | 34°13.89' N | 24° 49.84' E | 1000 | 2244* | 7 | 11 | 11 |
|  | 7 | 22/05/15 15:02 | 34° 12.61' N | 24° 50.76' E | 500 | 2886 | 8 | 11 | 11 |
|  | 8 | 22/05/15 17:04 | 34° 12.66' N | 24° 50.56' E | 2871 | 2886 | 0 | 11 | 11 |
| EAST IONIAN | 9 | 26/05/15 12:51 | 36° 41.84' N | 20° 07.32' E | 500 | 3175 | 8 | 11 | 11 |
|  | 10 | 26/05/15 14:44 | 36° 41.57' N | 20° 07.21' E | 3165 | 3175 | 0 | 11 | 11 |
| SOUTH TYRRHENIAN | 11 | 30/05/15 10:05 | 39° 10.43' N | 10° 53.47' E | 500 | 2812 | 8 | 11 | 11 |
|  | 12 | 30/05/15 13:36 | 39° 11.44' N | 10° 52.37' E | 2803 | 2812 | 0 | 11 | 11 |
| CENTRAL TYRRHENIAN | 13 | 31/05/15 05:21 | 40° 45.22' N | 11° 30.28' E | 500 | 2466 | 8 | 11 | 11 |
|  | 14 | 31/05/15 07:14 | 40° 45.87' N | 11° 30.66' E | 2456 | 2466 | 0 | 11 | 11 |

Table 1: Station summary. For bottom depth, values with asterisk indicate that the measurement has been obtained from the vessel's echo-sounder whether than the altimeter interfaced to the CTD unit.



| MISSION ID | ARGO WMO | FIRST PROFILE ID | INTER-DISTANCE (km) | INTER-DURATION (h) | TEMP OFFSET (°C) | PSAL OFFEST | OPTODE SLOPE | OPTODE OFFSET ($\mu$mol.kg$^{-1}$) | FLUO N | FLUO R$^2$ | FLUO OFFSET (mg.m$^{-3}$) | FLUO SLOPE | SUNA SLOPE | SUNA OFFSET ($\mu$mol.L$^{-1}$) |
|---|---|---|---|---|---|---|---|---|---|---|---|---|---|---|
| LOVBIO083D | 6901764 | BCN | 0 | 0 | 0.0059 | 0.0150 | 0.9796 | 11.56 | 7 | 0.98 | 0.01 | 0.67 | 1.00 | 3.20 |
| LOVBIO084D | 6901765 | 000 | 1 | 19 | 0.0003 | 0.0031 | 1.0660 | 3.26 | 8 | 0.77 | 0.04 | 0.62 | 1.00 | 4.00 |
| LOVBIO085D | 6901766 | BCN | 0 | 0 | 0.0053 | 0.0081 | 1.0275 | 6.30 | 7 | 0.98 | -0.02 | 0.65 | 1.11 | 0.80 |
| LOVBIO086D | 6901767 | 000 | 3 | 7 | | | | | 8 | 0.86 | 0.03 | 0.49 | 1.00 | -2.80 |
| LOVBIO086D | 6901767 | 001 | 3 | 29 | 0.0021 | -0.0009 | 1.1045 | -3.59 | | | | | 1.00 | -2.70 |
| LOVBIO088D | 6901768 | 001 | 12 | 31 | 0.0052 | 0.0009 | 1.0235 | 6.66 | 8 | 0.89 | 0.04 | 0.63 | 1.00 | 2.10 |
| LOVBIO089D | 6901769 | 000 | 2 | 26 | 0.0214 | 0.0050 | 1.1626 | -14.87 | 8 | 0.82 | 0.03 | 0.58 | 1.00 | 3.90 |
| LOVBIO091D | 6901771 | 000 | 2 | 21 | 0.0085 | 0.0042 | 1.0658 | 0.51 | 8 | 0.93 | 0.02 | 0.55 | 1.17 | 0.10 |
| LOVBIO093D | 6901773 | 000 | 3 | 22 | 0.0067 | 0.0070 | 1.0923 | -2.40 | 8 | 0.99 | 0.01 | 0.51 | | |
| AVERAGE | | | 3 | 17 | 0.0069 | 0.0053 | 1.0652 | 0.93 | | | 0.02 | 0.59 | 1.04 | 1.08 |
| STD | | | 4 | 12 | 0.0064 | 0.0049 | 0.0564 | 8.12 | | | 0.02 | 0.07 | 0.07 | 2.74 |

**Table 2: BGC-Argo float summary. For every BGC-Argo float deployed with a CTD cast of reference, the distance and duration with the first profile of the float is indicated. The results of metrological verification by parameter is reported. STD stands for standard deviation.**





| Cast | Depth (m) | BT distance (m) | LDEO parameters | Error velocity (cm.s⁻¹) | | | Misfits L ms S (cm.s⁻¹) | Comments |
|---|---|---|---|---|---|---|---|---|
| | | | | L without S constraint | L with S constraint | S | | |
| 1 | 1721 | 26 | L+S+BT | 2.5 | 2.5 | 5.5 | 1.8 | |
| 2 | 498 | | L+S | 3.4 | 3.4 | 5.7 | 3.8 | |
| 3 | 2990 | 53 | L+S+BT | 20.3 | 18.9 | 6.5 | 19.2 | rough sea, high tilt |
| 4 | 501 | | L+S | 3.1 | 2.3 | 6.9 | 3.1 | 5 |
| 5 | 1243 | | L+S | 2.9 | 3.4 | 6.4 | 6.9 | |
| 6 | 996 | | L+S | 2.5 | 2.6 | 5 | 2.0 | |
| 7 | 496 | | L+S | 2.5 | 2.4 | 4.6 | 2.2 | |
| 8 | 2871 | 16 | L+S+BT | 2.9 | 4.8 | 4.7 | 6.1 | |
| 9 | 502 | | downlooker+S | 2.2 | 3.1 | 4.5 | 2.6 | uplooker failed, low battery |
| 10 | 3165 | 17 | downlooker+S+BT | 20.7 | 50.4 | 5.2 | 36.0 | |
| 11 | 497 | | L+S | 3 | 3 | 5.9 | 4.5 | 10 |
| 12 | 2805 | 7 | L+S+BT | 5.4 | 4.2 | 6.1 | 4.4 | |
| 13 | 505 | | L+S | 2.6 | 2.5 | 5 | 2.5 | |
| 14 | 2456 | 12 | L+S+BT | 2.8 | 2.8 | 5.3 | 3.6 | |

15 Table 3. Summary of ocean current profiles collected at the stations. Depth and bottom track (BT) distance, when available, are indicated. Processing details are provided with the type of data used: LADCP (L) data, SADCP (S) data, BT data. Statistics per profile on error velocity are reported as well as the misfits between LACDP and SADCP currents.



|  | Without SADCP constraint | With SADCP constraint |
|---|---|---|
| Error velocity (in cm.s$^{-1}$) | 2.9 | 3.1 |
| LADCP ms SADCP module (in cm.s$^{-1}$) | -0.94 +/- 3.1 | 0.17 +/- 1.1 |
| LADCP ms SADCP azimuth (in degree) | 5.4 +/- 38 | -0.02 +/- 23 |

**Table 4. Median values of uncertainties computed over the 800 ensembles collected during the 12 acceptable casts (without cast 3 nor cast 10).**

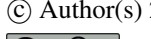



| CAST | TSG T | CTD T | TSG S | CTD S | TSG-CTD T | TSG-CTD S |
|---|---|---|---|---|---|---|
| 2 | 18.874 | 18.865 | 38.861 | 38.866 | 0.009 | -0.005 |
| 3 | 18.878 | 18.868 | 38.863 | 38.864 | 0.010 | -0.001 |
| 4 | 21.444 | 21.427 | 38.817 | 38.819 | 0.017 | -0.002 |
| 5 | 21.404 | 21.405 | 38.807 | 38.809 | -0.001 | -0.002 |
| 6 | 20.360 | 20.281 | 39.170 | 39.174 | 0.079 | -0.004 |
| 7 | 20.060 | 20.205 | 39.175 | 39.179 | -0.145 | -0.004 |
| 8 | 20.956 | 20.618 | 39.174 | 39.169 | 0.338 | 0.005 |
| 9 | 20.364 | 20.347 | 38.472 | 38.485 | 0.017 | -0.013 |
| 10 | 20.334 | 20.339 | 38.494 | 38.528 | -0.005 | -0.034 |
| 11 | 20.185 | 20.177 | 38.023 | 38.029 | 0.008 | -0.006 |
| 12 | 20.001 | 20.155 | 38.021 | 38.030 | -0.154 | -0.009 |
| 13 | 20.081 | 20.070 | 37.909 | 37.915 | 0.011 | -0.006 |
| 14 | 20.105 | 20.053 | 37.901 | 37.909 | 0.052 | -0.008 |
|  |  |  |  | AVE | 0.020 | -0.007 |
|  |  |  |  | STD | 0.122 | 0.009 |

**Table 5: Temperature and salinity measured during stations by the TSG system and the CTD unit. The misfit is computed by parameters and by cast. An average value with its standard deviation is indicated. STD stands for standard deviation.**



| Batch | Date of the batch | Thiosulfate reagent batch | Blank volume (mL) | STD (Blank volume) | Thiosulfate molarity (mol.L$^{-1}$ 20°C) | STD (Thiosulfate molarity) |
|---|---|---|---|---|---|---|
| 1 | 2015/05/13 | 1 | 0.0077 | 0.0016 | 0.01941 | 0.00004 |
| 2 | 2015/05/17 | 1 | 0.0060 | 0.0021 | 0.01940 | 0.00011 |
| 3 | 2015/05/22 | 1 | 0.0099 | 0.0025 | 0.01948 | 0.00005 |
| 4 | 2015/05/23 | 1 | 0.0079 | 0.0032 | 0.01946 | 0.00008 |
| 5 | 2015/05/27 | 1 | 0.0063 | 0.0022 | 0.01948 | 0.00004 |
| 6 | 2015/05/30 | 2 | 0.0070 | 0.0016 | 0.01950 | 0.00006 |
| 7 | 2015/05/31 | 2 | 0.0070 | 0.0027 | 0.01947 | 0.00010 |

**Table 6: Temporal evolution of reagent blank and the thiosulfate concentration during the cruise. STD stands for standard deviation.**





| Cast | Bottle | Pressure | Salinity | O$_2$ (µmol.kg$^{-1}$) | STD (O$_2$) | Replicates |
|------|--------|----------|----------|------------------------|-------------|------------|
| 12   | 1      | 2844     | 38.500   | 190.7                  | 0.40        | 6          |
| 14   | 11     | 10       | 37.895   | 226.5                  | 0.28        | 5          |

**Table 7: Results from reproducibility exercises for the dissolved oxygen samples withdrawn from the same Niskin bottle.**

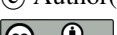


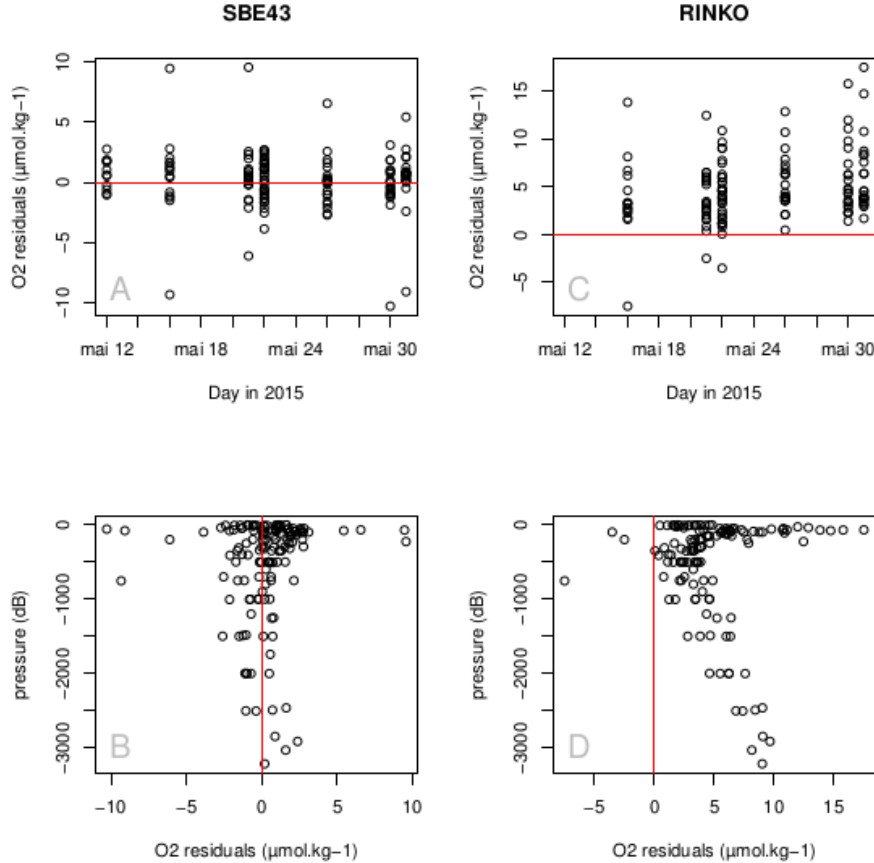

**Figure 2: Oxygen residuals between sensor and Winkler measurements, plotted in function of time (upper panels) and in function
of depth (lower panels). The residuals for the electrochemical sensor are plotted in the left panels, the ones for the optode in the right
panels.**



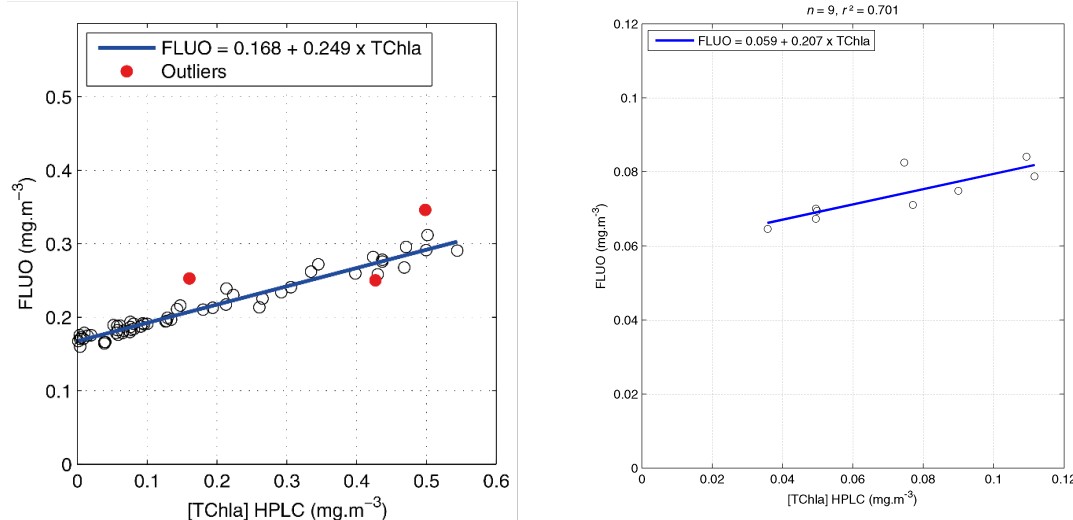

**Figure 3: Linear models for calibration of the Aqua Tracka III fluorometer (CTD unit, left panel) and of the Turner fluorometer (TSG system, right panel). The equations of the regression lines (in blue) are indicated.**



| Pigment | Variable name | Units | detection wavelength (nm) | Limit of Detection ng/inj | Limit of Detection for 2L filtered (in mg.m$^{-3}$) |
|---|---|---|---|---|---|
| Chlorophyll c3 | CHLC3 | mg.m$^{-3}$ | 450 | 0.015 | 0.0002 |
| Chlorophyll c1+c2 | CHLC2 | mg.m$^{-3}$ | 450 | 0.018 | 0.0002 |
| Sum Chlorophyllide a | CHLDA | mg.m$^{-3}$ | 667 | 0.016 | 0.0002 |
| Peridinin | PERI | mg.m$^{-3}$ | 450 | 0.007 | 0.0001 |
| Sum Phaeophorbid a | PHDA | mg.m$^{-3}$ | 667 | 0.009 | 0.0001 |
| 19'-Butanoyloxyfucoxanthin | BUT | mg.m$^{-3}$ | 450 | 0.009 | 0.0001 |
| Fucoxanthin | FUCO | mg.m$^{-3}$ | 450 | 0.009 | 0.0001 |
| Neoxanthin | NEO | mg.m$^{-3}$ | 450 | 0.009 | 0.0001 |
| Prasinoxanthin | PRAS | mg.m$^{-3}$ | 450 | 0.009 | 0.0001 |
| Violaxanthin | VIOLA | mg.m$^{-3}$ | 450 | 0.012 | 0.0001 |
| 19'-Hexanoyloxyfucoxanthin | HEX | mg.m$^{-3}$ | 450 | 0.009 | 0.0001 |
| Diadinoxanthin | DIADINO | mg.m$^{-3}$ | 450 | 0.014 | 0.0002 |
| Alloxanthin | ALLO | mg.m$^{-3}$ | 450 | 0.015 | 0.0002 |
| Diatoxanthin | DIATO | mg.m$^{-3}$ | 450 | 0.015 | 0.0002 |
| Zeaxanthin | ZEA | mg.m$^{-3}$ | 450 | 0.014 | 0.0002 |
| Lutein | LUT | mg.m$^{-3}$ | 450 | 0.014 | 0.0002 |
| Bacteriochlorophyll a | BCHLA | mg.m$^{-3}$ | 770 | 0.010 | 0.0001 |
| Divinyl Chlorophyll b | DVCHLB | mg.m$^{-3}$ | 450 | 0.004 | 0.0001 |
| Chlorophyll b | CHLB | mg.m$^{-3}$ | 450 | 0.004 | 0.0001 |
| Total Chlorophyll b | TCHLB | mg.m$^{-3}$ | 450 | 0.004 | 0.0001 |
| Divinyl Chlorophyll a | DVCHLA | mg.m$^{-3}$ | 667 | 0.011 | 0.0001 |
| Chlorophyll-a | CHLA | mg.m$^{-3}$ | 667 | 0.011 | 0.0001 |
| Total Chlorophyll a | TCHLA | mg.m$^{-3}$ | 667 | 0.011 | 0.0001 |
| sum Phaeophytin a | PHYTNA | mg.m$^{-3}$ | 667 | 0.007 | 0.0001 |
| Sum Carotenes | TCAR | mg.m$^{-3}$ | 450 | 0.013 | 0.0002 |

**Table 8: list of parameters in the pigment dataset, the name of variable, the units, and for each pigment, the detection wavelengths and the associated limits of detection in ng per injection.**

