# Peer review of "Hydrography in the Mediterranean Sea during a cruise with RV Tethys 2 in May 2015"

_Earth System Science Data, 2017_

## Referee Comment (RC1) · Anonymous Referee #1 · 16 Jan 2018

**General Comments**

Dear Editor.

I have read the manuscript entitled "Hydrography in the Mediterranean Sea during a cruise with RV Tethys 2 in May 2015" by Taillandier et. al. and I have also browsed the accompanying Datasets.

The manuscript provides an in-depth description of a data set collected within a cruise specifically planned to provide an assessment of the initial state of a set of biogeochemical Argo floats deployed within the cruise. The record addressed here corresponds to the cruise acquisition from CTD and underway systems, including hydrography, current

velocities and biogeochemical variables.

Data format is understandable, adequate for a spreadsheet thought not following standards as e.g. SeaDataNet nor using a Hierarchical Data Format to ease the direct access from users. This forces the user to implement its own code to import the dataset into analysis software.

My main concern about the paper is that there is an extensive description of data collection and processing procedures (too detailed in my view), while the presentation of the record is almost absent. For each variable group the authors provide the same sequence of subsections: Presentation of the different measurements → Data Processing → Data quality Control. Most of what it is shown directly replicates the standard procedures that can be found on available documents, as the GO-SHIP Manual or technical notes related to equipments. It is even provided the Seabird software sequence applied for the CTDs (as recommended for their CTD model/sensors).

I would suggest the authors to reduce the details on processing whenever it follows any published protocol, and also reduce the details of calibration coefficients/parameters determination, or at least try to provide the information graphically. It would be desirable that data gathering techniques and processing procedures could be audited by external reviewers to such a high level of detail, but this would be a very technical task. It is unfeasible for a science reviewer to check that procedures for all sensors and analysis from samples follow good practices. Note that due to the different disciplines involved the ms imply 15 co-authors; surely all of them specialized in few of the presented measurements. For example I cannot assess whether the nutrient values provided are meaningful or not.

As I understand, the aim of the ESSD paper should be providing only the processing details that, for whatever reason, do not follow the published standards, and to show the dataset highlighting its significance with respect to the existing record. Thus, the potential user do not have to download and plot himself the record before knowing

whether it will be useful for his purposes.

Also, I acknowledge that this BGC-Argo pre-launch exercise is interesting but I am not sure that this single cruise is enough to support a ESSD paper on its own. I have browsed oceanographic cruise data papers in the ESSD journal and found few, typically covering very extensive campaigns (e.g. doi:10.5194/essd-7-231-2015). The dataset presented here provide profiles just at 7 sites. I feel that data papers are more adequate for collections of cruises. This appreciation is of course subjective and, as long as the editor considers the contents of this dataset enough to support a paper, I will not object.

Therefore, in case that the editor considers the dataset present gathered is sufficient for a single ESSD paper, I suggest a major revision. The revised draft should substantially reduce the details on processing and avoid most tables (up to 8 in the current version) and include figures of the measured fields (CTD profiles, LADCP profiles, biogeochemical) so readers can see spatial changes across different basins. SADCP record should also be shown as an oceanographic section. I also miss extended discussion on the role of short-term variability in the misfit between CTD profiles and Argo floats (e.g. is the misfit in Argo vs CTD hydrography comparable for example to that seen among consecutive CTDs at the same site?). Finally, it would be interesting to see (graphically) how nutrients and O2 at depth relate to known background climatological fields.

**Specific Comments**

p2. l.4-5. Provide references on these "few ocean observation sites".

Section 1.2 I understand that the cruise is dedicated to the maintenance of the Argo-BGC fleet by adding more floats to the array. Some questions arise to me:

- Can you provide info on how many buoys were active at the time of the cruise?.

- Was not possible to perform any calibration profile at positions of active Argo-BGC at the time of the cruise?.

- Where were the 10 floats deployed? I assume 7 were launched at the CTD positions provided but what about the remaining three?, do these correspond to the Levantine basin?.

p3. l.20. Why only the LOV BGC-Argo were considered?

p3. l.26. "Note that the floats were programmed to profile everyday at noon". Should be pointed before, readers not familiar with this array expect Argo floats to profile once every 10 days. Moreover you have said that the floats cycle on weekly basis (p.2 l.8) and again say in the conclusions.

p3. para6. Please explain better, didn't you get permission to perform CTDs in the Levantine basin but to sail there and launch Argo floats?

Section 2.1.2. The reference Le Bot et. al. is absent in the bibliography. Could you explain differences (if any) between the Cascade processing and the more widespread software CODAS? Could you comment further in the bottom track correction through Gebco?. It is not clear to me whether you use LADCP downlooker bottom track to constrain the final profile solution.

p7. l.22– explain further why choosing the layer 750–1000 m, how does the de-correlation scale increase with depth? Is 750 m deeper than the extent of a typical eddy?. l.25, what is the $0.01°C$ (or in PS) threshold?

p9. l.15 Regarding the O2 sensor hysteresis correction, are you following any Sbe technical note directly (if so please indicate which version) or did you programmed specific software?.

l.21 why 2.8 standard deviation?.

l.25 Why do you use CTD temp instead of the built-in temperature of Rinko software?.
You cannot account for inner sensor thermal lag using an external thermometer.

p.15 l. 24– I do think that a Dataset including pre and post mission calibration would be very valuable.

p.22 (Table 2). Provide deployment positions.

p.23 (Table 3). How is the error magnitude compared to the actual velocity?.

p.28 (Fig.2). Should not the residuals of the Rinko sensor provide zero-average bias after the calibration against Winkler?.

**Technical Corrections**

- p2. l.30 "are to be" sounds weird to me. Next sentence delete "and".

- p3. l.2. 3000nm, add space. Also along para.3 (1000m etc)

- p3. l.25. Re-word "calibration exercises could have been drawn".

- p4. l.26 I would say assembled instead composed.

- p8. l.1. The date in ref to Winkler work does not match that on the bibliographic list.

- p12. l.31 should read right panel.

- p15. l.20 replace "until" by "up to"

---

## Editor Comment (EC1) · G. M. R. Manzella (Editor) · 22 Jan 2018

Comments to: Hydrography in the Mediterranean Sea during a cruise with RV Tethys 2 in May 2015, by Vincent Taillandier1 et al.

The paper is a report on data from an oceanographic cruise fully dedicated to the maintenance and the metrological verification of a biogeochemical observing system based on a fleet of BGC-Argo floats. The objective of the cruise is very important and the details on the quality assessment and control procedures are too much accurate. However, the paper lacks important information on the existing knowledge on - biochemistry in the Mediterranean and changes - historical data and gaps in observing systems Such information should partially justify the important efforts done during the

Tethys cruise. There is another aspect that should be included in the paper: are there seasonal changes? Can a single cruise provide useful information to assess seasonal variabilities? The paper is going to much in details on qc procedures, that can be found in other publications and reports and is not providing any information on the contribution to the scientific knowledge on the Mediterranean ecosystems.

As it is now, the paper has be strongly revised.

---

## Referee Comment (RC2) · Anonymous Referee #2 · 29 Jan 2018

**Review of the paper entitled:**

Hydrography in the Mediterranean Sea during a cruise with RV Tethys 2 in May 2015

V. Taillandier et al.

**Manuscript essd-2017-119**

The manuscript reports about hydrographical and biogeochemical data which were gained in the Mediterranean Sea during a cruise with the research vessel Tethys 2 in May 2015. The aim of the cruise was to calibrate BGC –Argo float data and to introduce procedures for this kind of calibration. The description of observation methods and data processing is clear and detailed (maybe sometimes even too detailed for those observation methods which are standard procedures). The data and presentation quality is good and the data is accessible under the corresponding doi number. To my opinion the manuscript is worthwhile to be published in Earth System Science Data with some minor corrections discussed below.

1. Title: could be improved. For example: Hydrography and biogeochemics in the Mediterranean Sea during a cruise with RV Tethys 2 in May 2015 to calibrate BGC-Argo floats
2. Page 1, line 25: for (instead of to) temperature …?
3. Page 3, line 8-13: this is standard for a research vessel, can be shortened.
4. Page 3, line 15-16: I understand that usually two casts were taken at a station. But what about samples? Were they taken at standard depths? Or, different depths at different stations?
5. Page 5, line 4: When CTD profile available … were available
6. Page 5, line 8-14: I am not sure if I got it right: the transects had to be done in order to correct the misalignment angle of the ADCP? If so, you should use i.e. the CODAS software (Hawaii) for ADCP data analysis, this calculation is included there.
7. Page 7, paragraph 2.2.3: just a comment: it is uncommon, not to check especially salinity against samples, also in case of the TSG. You were lucky that your sensors remained stable. In case not, you would not be able to reproduce the station when it happened and to correct values accordingly.
8. Page 8, line 1: I guess it is 1988 instead of 1888.
9. Page 8, 1-30: to my opinion too detailed, it is standard method
10. Page 12, line 23 instead of upper panel, left panel?
11. Table 3 is confusing: in LDEO you list the components but in the following columns you calculate it with and without these components? It's a contradiction?

---

## Author Comment (AC1) · 14 Feb 2018

**Manuscript ESSD-2017-119 – Responses to reviewers**

We thank the reviewers and the associate editor for their thorough reading of the manuscript and for their constructive criticisms. We have modified the paper following their suggestions and we believe that the manuscript has improved thanks to them.

Our responses are detailed in the following, addressing all the points raised by the reviewers. The responses hereinafter and the corresponding changes in the revised manuscript appear in blue for convenience.

**Associate editor:**

The paper lacks important information on the existing knowledge on (i) biochemistry in the Mediterranean and changes (ii) historical data and gaps in observing systems. Such information should partially justify the important efforts done during the Tethys cruise.

*The BGC-Argo program has been developed in the context of stock assessments of phytoplankton biomass over oceanic basins, and the underlying changes at seasonal scales and under increasing anthropic perturbations. We believe that these aspects have been described in the Section 1.1, with an overview of the biogeochemistry in the Mediterranean (the typical functioning, the large-scale gradients, the different trophic regimes distributed as bio-provinces). As also suggested by the reviewer #1, the dataset is now presented in the context of this existing knowledge, see Section 4 in the revised manuscript.*

*Concerning observing systems, the Section 1.1 provides elements of historical archives (mainly ocean color products, time-series at observation sites as Dyfamed); it is stressed how the gaps of this observing system (mainly the vertical structuration of biomass at seasonal scales) can be filled by BGC-Argo data.*

There is another aspect that should be included in the paper: are there seasonal changes? Can a single cruise provide useful information to assess seasonal variabilities?

*Seasonal changes on biogeochemistry in the Mediterranean Sea have been studied from satellite ocean color records as mentioned in Section 1.1. Different trophic regimes have been characterized from seasonal cycles of surface chlorophyll-a concentration. This bio-regionalization revealed that a single cruise can provide useful information on spatial distributions (bioregions) but would need more intensive surveys to catch the seasonal signals. This is the motivation of this association of a cruise for reference data and time extension from BGC-Argo floats. These elements appear in the Section 4 of the revised manuscript.*

The paper is going to much in details on qc procedures, that can be found in other publications and reports and is not providing any information on the contribution to the scientific knowledge on the Mediterranean ecosystems.

*This comment has also been pointed out by the two reviewers and a major revision of the manuscript has been made to reduce information on qc procedures. Changes are detailed hereinafter in the responses to the reviewers' comments.*

**Reviewer #1:**

Data format is understandable, adequate for a spreadsheet thought not following standards as e.g. SeaDataNet nor using a Hierarchical Data Format to ease the direct access from users. This forces the user to implement its own code to import the dataset into analysis software.

*The data format consists in tables constructed as comma separated values, archived in Ascii files, with a full description of the data collection. This choice has been suggested by the data manager that attributed the doi number to this dataset, in order not to restrict to any specific analysis software. This data format has not been modified, also following the appreciation of the reviewer #2.*

My main concern about the paper is that there is an extensive description of data collection and processing procedures (too detailed in my view), while the presentation of the record is almost absent.

*We have followed the recommendation of the reviewer: in the revised manuscript, the description of the processing procedures and qc protocols have been simplified (important cuts in the text of Section 2 and four tables suppressed) whereas the presentation of the record has been developed (two new figures and a new paragraph in Section 4). Details on the underlying modifications are drawn hereinafter.*

I would suggest the authors to reduce the details on processing whenever it follows any published protocol, and also reduce the details of calibration coefficients/parameters determination.

*Major changes have been achieved in the Section 2 in order to simplify the description of the sensing methods and QC protocols. Here is a detailed list of the changes achieved in the revised manuscript.*

*Section 2.2.1: details on deployment procedure of deployment for CTD underwater unit have been removed as it followed the standard procedure of the GO-SHIP manual.*

*Section 2.2.2: details on data processing for TSG and CTD profiles have been removed (and references therein) as they followed the standard procedure of the GO-SHIP manual.*

*Section 2.3.1: details on the Winkler method have been removed as they followed the recommendations of Langdon (2010). The description of the functioning for the electrochemical sensor as well as the optode sensors (and references therein) has been simplified.*

*Section 2.3.2: details on the Winkler protocol have been removed as they followed the recommendations of Langdon (2010).*

*Section 2.4.2: details on the linear regression model (and references therein) have been removed. Description of data processing for TSG fluorescence has been simplified.*

*Section 2.4.3: The quality control of HPLC data has been simplified as it followed the recommendations of Ras et al. (2008).*

*Section 2.5.1: details on nutrient sampling and titration (and reference therein) have been removed as they followed the recommendations of Kirkwood (1992) and Aminot and Kerouel (2007). Details on nitrate optical sensing (and references therein) have been removed.*

*Section 2.5.2: details on the data processing algorithm have been removed as they are reported in Paqueron et al. (2015).*

The revised draft should substantially reduce the details on processing and avoid most tables (up to 8 in the current version) …

*Following the reviewer's recommendation, four tables have been removed in the revised manuscript.*

*Section 2.1.3: the Table 4 has been suppressed accordingly and a sentence reporting the assessment has been added.*

*Section 2.2.3: the Table 5 has been removed and only the regression coefficient have been reported in the text.*

*Section 2.3.2: the Tables 6 and 7 have been removed as the description of the processing procedure has been simplified.*

*Section 2.4.2: the Figure 3 has been removed as it provided redundant information with the text. The Table 8 has been renamed as Table 4.*

… and include figures of the measured fields (CTD profiles, LADCP profiles, biogeochemical) so readers can see spatial changes across different basins. SADCP record should also be shown as an

oceanographic section.

*A new paragraph has been added at the beginning of the Section 4 of the revised manuscript. The record is presented in the frame of existing knowledge of the Mediterranean hydrology and biogeochemistry, in order to introduce the complementarity between the cruise and BGC-Argo. As suggested by the reviewer, two figures have been added for this purpose: a SADCP section in Figure 3, TS diagrams from CTD casts in Figure 4 (upper left panel), oxygen concentration profiles from CTD casts in Figure 4 (lower left panel), total chlorophyll-a concentration and nitrate concentration from water samples in Figure 4 (right panels).*

I also miss extended discussion on the role of short-term variability in the misfit between CTD profiles and Argo floats (e.g. is the misfit in Argo vs CTD hydrography comparable for example to that seen among consecutive CTDs at the same site?).

*The reviewer is right to underline that short-term variability can affect the comparison between CTD and BGC-Argo profiles, introducing natural variability in the measured misfit. This point will be discussed hereinafter in the response of a specific comment in the case of hydrology and on the mesoscale effects. The issue is even more challenging for some biogeochemical variables, that have signal on surface affected by diurnal cycle. We believe that a perfect concomitancy (float clamped in the CTD-Carousel) is the only way to characterize possible calibration shifts at deployment, but our ancillary dataset is still too small to provide a quantitative answer to this point. A sentence has been added in Section 4 to mention the reviewer's comment in the revised manuscript.*

Finally, it would be interesting to see (graphically) how nutrients and O2 at depth relate to known background climatological fields.

*The suggestion of the reviewer is fair, as soon as our report provides verification and quality control of the records by the comparison of several in-situ sensing methods rather than comparing them to climatology. Following the reviewer's comment, the dataset of oxygen and nitrate concentration has been replaced among previous large-scale field surveys (BOUM in 2009 and M83/4 in 2011). A sentence has been added in Section 4, first paragraph, of the revised manuscript.*

p2. l.4-5. Provide references on these "few ocean observation sites".

*Done*

Section 1.2. I understand that the cruise is dedicated to the maintenance of the Argo- BGC fleet by adding more floats to the array. Some questions arise to me: Can you provide info on how many buoys were active at the time of the cruise?

*At the time of the cruise there were 12 active BGC-Argo floats. This information has been added in the revised manuscript.*

Was not possible to perform any calibration profile at positions of active Argo-BGC at the time of the cruise?

*Among the active floats, 4 have been recovered and renewed during the cruise. This allowed to provide a post-mission calibration for these floats. The other active floats could not be reached given the available ship time and the constraints of regular port calls; their mission finished within the year without recovery.*

Where were the 10 floats deployed? I assume 7 were launched at the CTD positions provided but what about the remaining three? Do these correspond to the Levantine basin?

*The position of the 9 floats concerned by calibration exercises has been added in the Table 2. As explained in Section 1.2, we have encountered problems for working clearance in the Levantine basin.*

*Thus one float has been deployed without CTD cast (it does not appear in Table 2), and two other floats have been deployed in another location after calibration exercise at a CTD station (they appear in Table 2 with a star).*

p3. l.20. Why only the LOV BGC-Argo were considered?

*This sentence has been rewritten in the revised manuscript: we were meaning that only BGC-Argo floats were considered as several physical Argo floats have been also deployed during the cruise.*

p3. l.26. "Note that the floats were programmed to profile everyday at noon". Should be pointed before, readers not familiar with this array expect Argo floats to profile once every 10 days. Moreover you have said that the floats cycle on weekly basis (p.2 l.8) and again say in the conclusions.

*The cycling rate can be changed during missions thanks to Iridium communications (detailed in Leymarie et al., 2013). The current cycling period is on a weekly basis; however a daily cycling has used at the beginning of the mission in order to increase concomitancy with the CTD cast. This point has been clarified in the revised manuscript.*

p3. para6. Please explain better, didn't you get permission to perform CTDs in the Levantine basin but to sail there and launch Argo floats?

*We have got working clearance from the Greek authorities to perform CTD casts in the Levantine basin. However, a Turkish warship surveying this area during our cruise did not allow us to perform these CTD casts.*

Section 2.1.2. The reference Le Bot et. al. is absent in the bibliography. Could you explain differences (if any) between the Cascade processing and the more widespread software CODAS?

*CASCADE is the software used in routine to process VM-ADCP data collected in the French RV fleet. The principles of the two softwares are the same: edit and correct doubtful profiles, integrate navigation data in the processing. As underlined by the reviewer #2, CODAS proposes more options for the calibration aspects of the ADCP, in particular the calculation of misalignment angle of the instrument that would have prevented the L-shape calibration exercise performed during the cruise. A sentence has been added in section 2.1.3 and the reference of Le Bot et al. has been added to the revised manuscript.*

Could you comment further in the bottom track correction through Gebco?

*The reviewer is right to ask for clarification: actually no correction has been performed, only a mask on bottom detections using Gebco has been applied. The manuscript has been revised in agreement.*

It is not clear to me whether you use LADCP downlooker bottom track to constrain the final profile solution.

*The information if whether the bottom track is used or not in the final computation only appears in Table 3. For sake of reducing the processing details, the manuscript has not been modified according to this point.*

p7. l.22– explain further why choosing the layer 750–1000 m, how does the de- correlation scale increase with depth? Is 750 m deeper than the extent of a typical eddy?

*This is a difficult point to determine whether natural variability or instrumental shift cause misfits between CTD and float profiles. The main motivation of choosing the layer 750-1000 m is to compare characteristics of stable water masses (under the LIW) that should be less influenced by short scale variability of surface layer dynamics. The last paragraph of Section 2.2.3 has been rewritten according to the reviewer's comment in the revised manuscript.*

*Considering mesoscale variability (that can extend under 750 m), the distance between CTD and float*

*profiles remain short (few kilometers) with respect to the size of typical eddies (60 km). Moreover, the duration between profiles (less than 2 days) should remain short enough with respect to the displacement of typical eddies (some kilometers per day) to consider that the two profiles would keep influenced equally by mesoscale effects. A sentence has been added in Section 4 to mention the reviewer's comment.*

l.25, what is the 0.01°C (or in PS) threshold?

*The accuracy of 0.01 referred to Argo standards for uncertainties of measurements. This notion of threshold, unclear in this context, has been removed in the revised manuscript.*

p9. l.15 Regarding the O2 sensor hysteresis correction, are you following any Sbe technical note directly (if so please indicate which version) or did you programmed specific software?

*The SBE technical note (AN64-2) has been followed directly to modify the algorithm of Owens and Millard (1985). The reference has been added in the revised manuscript.*

l.21 why 2.8 standard deviation?

*The outliers are classically edited by comparing individual misfits to the overall standard deviation, that requires the specification of a threshold value. In our processing the threshold value of 2.8 standard deviation was set to edit outliers.*

l.25 Why do you use CTD temp instead of the built-in temperature of Rinko software? You cannot account for inner sensor thermal lag using an external thermometer.

*Considering optodes, built-in temperature is measured by an external sensor as there is no plumbing inside the sensor. We believe that the temperature at the location of the foil, used to derive dissolved oxygen concentrations by the Rinko software, would be the seawater temperature as effects of thermal cell inertia should not be significant compared to the accuracy and dynamical responses of built-in temperature sensors.*

p.15 l. 24– I do think that a Dataset including pre and post mission calibration would be very valuable.

*We agree with the reviewer: the records of this cruise provide valuable ancillary data for the BGC-Argo missions. They are archived to perform and distribute to the users community delayed mode adjustments in the time series of the BGC-Argo floats. A sentence has been added in Section 4 of the revised manuscript.*

p.22 (Table 2). Provide deployment positions.

*The Table 2 has been modified following the reviewer's comment. Only WMO number has been kept and a column with station deployment has been added.*

p.23 (Table 3). How is the error magnitude compared to the actual velocity?

*We are not sure to understand this comment. LADCP data processing with the software LDEO provides horizontal velocity profiles together with associated uncertainties (error velocity magnitude) by minimizing the misfits to different constraints.*

p.28 (Fig.2). Should not the residuals of the Rinko sensor provide zero-average bias after the calibration against Winkler?

*The Figure 2 shows the residuals between the Rinko sensor and Winkler measurements, before calibration against Winkler. A pressure-dependent slope is reported in Section 2.3.3, however a post-cruise processing of the Rinko data cannot be achieved because the calibration coefficients of this sensor are not available. That is why the SBE43 data have been used rather than Rinko data in the final dataset. This is now clearly mentioned in the revised manuscript.*

p2. l.30 "are to be" sounds weird to me. Next sentence delete "and".

*Done*

p3. l.2. 3000nm, add space. Also along para.3 (1000m etc)

*Done*

p3. l.25. Re-word "calibration exercises could have been drawn".

*Done*

p4. l.26 I would say assembled instead composed.

*Done*

p8. l.1. The date in ref to Winkler work does not match that on the bibliographic list.

*Done*

p12. l.31 should read right panel.

*Done*

p15. l.20 replace "until" by "up to"

*Done*

**Reviewer #2:**

The description of observation methods and data processing is clear and detailed (maybe sometimes even too detailed for those observation methods which are standard procedures).

*The descriptions of standard procedures for sensing, processing and quality control have been reduced in the revised manuscript.*

The data and presentation quality is good and the data is accessible under the corresponding doi number.

*We followed the appreciation of the reviewer: the format for the archive of the dataset has not been modified.*

Title: could be improved. For example: Hydrography and biogeochemics in the Mediterranean Sea during a cruise with RV Tethys 2 in May 2015 to calibrate BGC-Argo floats

*We followed the suggestion of the reviewer.*

Page 1, line 25: for (instead of to) temperature ...?

*We did not find this typo.*

Page 3, line 8-13: this is standard for a research vessel, can be shortened.

*This paragraph has been cut in the revised version and main information moved to the following paragraph.*

Page 3, line 15-16: I understand that usually two casts were taken at a station. But what about samples? Were they taken at standard depths? Or, different depths at different stations?

*The deep cast was composed of only standard levels, whereas the sampling around the deep chlorophyll maximum was refined during the shallow cast. The sampling strategy is now detailed in the revised manuscript.*

Page 5, line 4: When CTD profile available ... were available

*Done*

Page 5, line 8-14: I am not sure if I got it right: the transects had to be done in order to correct the misalignment angle of the ADCP? If so, you should use i.e. the CODAS software (Hawaii) for ADCP data analysis, this calculation is included there.

*The reviewer is right: contrary to CASCADE software, CODAS software proposes the option to correct the misalignment angle of the ADCP, that would have prevented the L-shape calibration exercise performed during the cruise. A sentence has been added in section 2.1.3 of the revised manuscript.*

Page 7, paragraph 2.2.3: just a comment: it is uncommon, not to check especially salinity against samples, also in case of the TSG. You were lucky that your sensors remained stable. In case not, you would not be able to reproduce the station when it happened and to correct values accordingly.

*Thank you for this comment. This comparison takes less space in the revised manuscript while the Table 5 have been cut.*

Page 8, line 1: I guess it is 1988 instead of 1888.

*Even if we understand that it could appear to be strange, it is interesting to note that the original paper of the Winkler method has been published in 1888 and that this method has remained the standard method to this day with only marginal improvements that are taken into account in the recommendations of Langdon 2010. However, as mentioned by reviewer 1, in the reference list the publication of Winkler was incorrectly reported with the year 1988.*

Page 8, 1-30: to my opinion too detailed, it is standard method

*The Section 2.3.1 has been simplified and strongly reduced in the revised manuscript (see response to reviewer #1 for details).*

Page 12, line 23 instead of upper panel, left panel?

*The Figure 3 has been suppressed following the suggestion of the reviewer #1.*

Table 3 is confusing: in LDEO you list the components but in the following columns you calculate it with and without these components? It's a contradiction?

*For every casts, the error velocities were computed for three sets of profiles: one as measured by LADCP only, one as measured as SADCP only, and one as processed from LADCP measurements under SADCP constraint. In function of the results were chosen the final process parameters reported in the column. The order of the columns of the table and the table caption have been changed in the revised manuscript.*

**Hydrography and biogeochemistry dedicated to the Mediterranean BGC-Argo network 
[revised manuscript text omitted]
 (0-500 dbar) and one deep cast (0-bottom). Standard levels were chosen for the deep cast (bottom, 2000 dbar, 1500 dbar, 1250 dbar, 1000 dbar, 750 dbar, 500 dbar, salinity maximum, 200 dbar, chlorophyll maximum, 10 dbar). The shallow cast was composed of six standard levels (500 dbar, 200 dbar, 150 dbar, 50 dbar, 10 dbar, 5 dbar) and five levels dedicated to the sampling of the deep chlorophyll maximum. This sampling strategy has been reduced to a single cast (0-1000 dbar) in case of rough sea conditions, or extended with another cast (0-1000 dbar) for calibration purposes. The number of

15    casts and samples are summarized in Table 1, with a total of 60 pigment samples, 148 oxygen samples, and 154 nutrient samples.

The cruise was prepared in coordination with the Euro-Argo infrastructure so that series of Argo and BGC-Argo floats were provided by different European institutes (BSH Germany, OGS Italy, LOV France). Hereinafter, only the BGC-Argo component is considered. At the time of the cruise, there were twelve active floats; four of these floats have been recovered

20    and ten new floats have been deployed during the cruise. The standard way consisted in deploying BGC-Argo floats at the end of every stations, as listed in Table 2. Calibration exercises have been drawn assumed that the CTD casts and the first float profiles can be considered as co-located in time and space. That is why the floats were programmed to profile everyday at noon at the beginning of their mission. The first deep profile (0-1000 m) acquired by the floats could occur the day of the station if deployed early in the morning, or the day after if deployed later, as reported in Table 2.

[revised manuscript text omitted]

5    supposedly steady over the 2h duration of the exercise. The two transects were significantly different in amplitude and azimuth. Corrections on misalignment angle (1.1 degree), amplitude factor (1.004) and pitch thresholds (1 and 1.5 degree) for the SADCP have been proposed in order to reduce the misfits between transects. Note that this calibration exercise would have been prevented in case of using CODAS software that proposes the option to compute the misalignment angle of the SADCP. Quality controlled data set of ocean currents along ship track have been post-processed thanks to these corrections.

10    This post-processed SADCP dataset was also performed during stations in order to assess and improve the quality the 14 LADCP profiles. As reported in Table 3, all the profiles unless at casts 3 and cast 10 are characterized by low velocity errors and acceptable misfits with SADCP profiles. The median value of these uncertainties over the 12 acceptable casts using 1-second resolution profiles (approximately 800 ensembles) was evaluated to -0.94 +/- 3.1 cm.s$^{-1}$ in module and 5.4 +/- 38 degree in azimuth without the SADCP constraint. Under SADCP constraint the median value reaches to 0.17 +/- 1.1 cm.s$^{-1}$ in module

[revised manuscript text omitted]

5 Considering BGC-Argo floats, the comparison with CTD profiles was performed over the layer 750 dbar – 1000 dbar, where water mass characteristics would remain stable enough to ascribe misfits as instrumental calibration shifts whether than natural variability. The misfits between temperature measurements and practical salinity measurement at geopotential horizons were computed and median values provided for every BGC-Argo floats. The median offsets are reported in Table 2. They amplitudes remained of 0.01°C in temperature or 0.01 in practical salinity, unless in two cases. A large temperature offset stand for WMO

10 6901769. A large practical salinity offset was reported for WMO 6901765 however deployed in exact concomitancy with the CTD profile.

**2.3 Oxygen concentration**

**2.3.1 Presentation of the different measurements**

Concentration of dissolved dioxygen ($O_2$) in seawater, hereafter described as oxygen, was measured with three techniques: the

15 classical iodometric Winkler method, an electrochemical oxygen sensor, optical oxygen sensors.

Oxygen concentration was measured following the Winkler method (Winkler, 1888) with potentiometric endpoint detection (Oudot et al., 1988) on discrete samples collected with Niskin bottles. For sampling, reagents preparation and analysis, the recommendations from Langdon (2010) have been carefully followed.

Oxygen concentrations have been measured by a Seabird SBE43 (serial number 0587) electrochemical sensor interfaced with

20 the CTD unit. This sensor was plumbed in the pumped circuit following the GO-SHIP guidelines (Hood et al., 2010).

Oxygen optical measurements (also called optode measurements) were collected by two types of sensors. One Rinko III dissolved oxygen sensor from JFE Advanced Co. Japan (serial number 171) was interfaced with the CTD unit using the analog output voltage. Aandeeraa 4330 optodes were mounted on every BGC-Argo floats.

**2.3.2 Data processing**

25 The titration volumes were converted to oxygen concentrations in $\mu mol.kg^{-1}$ by following the calculation procedure proposed in Langdon (2010). The precision of the Winkler measurements was estimated by reproducibility tests based on 5 or 6 replicates for samples withdrawn from same Niskin bottles. The standard deviation on the replicate measurements was lower than 0.4 $\mu mol.kg^{-1}$.

The sensor signal of the SBE43 was aligned to temperature and pressure scans considering a unique plumbing configuration

30 for cruise, by an advance of 3s. The raw signal was then converted to an oxygen concentration with 13 calibration coefficients. The method is based on the Owens and Millard (1985) algorithm that has been slightly adapted by Seabird in the data

processing software using a hysteresis correction (Sea Bird Scientific, 2014). A new set of calibration coefficients for this sensor has been determined after the cruise, it has been used to post-process the whole dataset. Only three coefficients (the oxygen signal slope, the voltage at zero oxygen signal, the pressure correction factor) among the 13 determined by the pre-cruise factory calibration of the sensor were adjusted with the following procedure. The oxygen concentrations measured by

5    Winkler were matched with the signal measured by the sensor at the closing of the Niskin bottles. The three values were fitted by minimizing the sum of the square of the difference between Winkler oxygen and oxygen derived from sensor signal. Outliers were discarded when the residuals exceeded 2.8 standard deviation of the residuals until no more outliers remain.

The Rinko optode provided continuous voltage output at 24 Hz, which has been directly converted to an oxygen concentration with the Matlab code developed by the manufacturer. The original calibration coefficients have been used. To process the

10   results, the temperature measured from the CTD unit was preferred to the built-in temperature of the sensor.

The Aanderaa optodes 4330 output signal is a C1 raw phase (phase from the blue light excitation), a C2 raw phase (phase from the red light excitation), and the optode temperature. The calculation of oxygen concentrations from the optode signal follows the recommendations of Thierry et al. (2016). The calibrated phase estimated from the C1 and C2 raw phases is converted in oxygen concentration by the Stern-Volmer equation proposed by Uchida et al. (2008) using seven calibration coefficients (the

15   so-called Stern-Volmer-Uschida coefficients). The oxygen concentration is then corrected from salinity and pressure effects. The pressure compensation is estimated following Bittig et al. (2015) with a step of phase adjustment. Finally, concentrations are expressed in $\mu mol.kg^{-1}$ by using the potential density derived from the CTD measurements of BGC-Argo floats.

**2.3.3 Data quality Control**

Winkler measurements on discrete samples collected during upcasts were considered as the reference oxygen value because

20   they rely on a reference material ($KIO_3$ standard) given with a precision on replicate measurements lower than 0.4 $\mu mol.kg^{-1}$. The reference Winkler measurements were used to adjust the calibration coefficients of the CTD oxygen sensor (SBE43), as described behalf. The corrected oxygen profiles during downcasts from the SBE43 at stations were considered as the reference profile for optode measurements from BGC-Argo floats. This quality control was based on the downcasts profiles at 1 dbar resolution collected whether by the electrochemical sensor SBE43 or the optode RINKO.

25   Residuals with Winkler measurements were expressed as the difference on an isobaric horizon between the sensor oxygen and the Winkler oxygen. A sensor error was estimated as the root mean square error on the residuals. Results are reported in Figure 2, where the residuals over the entire cruise are plotted as a function of time and depth. Residuals appear higher and more variable in the upper part of the water column, most probably due to enhanced oxygen gradients and changes on isobaric horizons between downcasts and upcasts. For electrochemical measurements, no significant offset or drift were observed; the

30   sensor error over the entire cruise is 2.4 $\mu mol.kg^{-1}$. For RINKO optode measurements, the sensor error over the entire cruise was 6.0 $\mu mol.kg^{-1}$ and a systematic offset of 4.8 $\mu mol.kg^{-1}$ was observed. Moreover, a significant increase of the residuals with depth (0.0022 $\mu mol.kg^{-1}.dbar^{-1}$) was observed below 200 dbar. Thus, the SBE43 data have been used rather than the RINKO data in the final record.

[revised manuscript text omitted]

The BGC-Argo floats deployed during the cruise were transmitting the raw data of the SUNA (i.e. absorbance spectrum from 217 to 250 nm), which allowed a post-processing with the algorithm of Pasqueron de Fommervault et al. (2015). A spike test was applied as well as a test for saturation based on the raw absorption spectrum. Nitrate concentration data computed from a spectrum for which more than 25% of the channels saturate (i.e. reached the maximum value of numerical counts) were

5    discarded. This was the case of one BGC-Argo float (WMO6901773).

**2.5.3 Data quality control**

The SUNA sensors also undergo offset and gain (Johnson et al., 2013) that were corrected using as reference the measurements on discrete samples. Given that surface nitrate concentrations in May and June in the Mediterranean Sea stand below the limit of detection of the sensor (Pasqueron de Fommervault et al., 2015), an offset was computed as the difference between an

10   assumed surface concentration of zero and the mean nitrate value measured from 5 to 30m. A gain was then calculated with a match up between sensors measurements and nitrate concentration at discrete depths. Gain correction was applied only if the misfits between sensor derived and reference concentrations below 950 dbar did not exceed 10% of the deep reference value. The correction coefficients per BGC-Argo float are reported in Table 2. A slope of 1 was estimated for most of the cases, and the offsets ranged from -2.70 µmol.L$^{-1}$ to 3.90 µmol.L$^{-1}$.

15   ## 3 Data availability

The final data set concatenates the different collections during the cruise, which are vertical profiles and bottle samples at CTD casts, along track measurements at surface and at depth. This data set benefits for post-cruise corrections described in the previous sections. A unique convention has been used to identify bad data, absent data, or not reported data: they have been assigned to the value -999.

20   The quality control provided to discrete samples collection has been assigned with a quality flag. The quality code set up for WHP bottle parameters data has been used, in particular: "2: Acceptable measurement", "5: Not reported", "9: Sample not drawn for this measurement from this bottle".

Data are published by SEANOE operated by SISMER within the framework of the in the information system ODATIS. Data at stations are available under doi:10.17882/51678, data along ship track are available under doi:10.17882/51691.

25   ## 4 Discussion and conclusions

With an extension of about 25 degrees in longitude, this cruise covered the central Mediterranean Sea and part of its north western and eastern basins. High resolution ADCP data (Figure 3) reveals some well know patterns of the surface circulation in this area (the cyclonic gyre in the Ligurian basin, the eastward surface flow in the Levantine) as well as ubiquitous mesoscale activity. Seven stations were chosen in this transect (one in the Ligurian, two in the Tyrrhenian, two in the Ionian, two in the

Levantine) in order to provide a large-scale record on the hydrography and biogeochemistry over the Mediterranean Sea. As shown in Figure 4 (upper left panel), there is a clear separation of water masses characteristics between the eastern and western basin, with a clear longitudinal gradient as deep waters and intermediate waters become eastwards saltier and warmer. Associated to this water mass distribution, biogeochemical traits clearly come up with important differences between basins and a relative homogeneity among basins. As shown in Figure 4 (lower left panel), the oxygen minimum of the intermediate waters is the lowest in the western stations, and deep waters are more oxygenated in basins directly influenced by winter convection (Ligurian and Ionian). The nutrient distribution shows also the eastern depletion of nitrates in deep waters, shallower nitraclines in the western basin, and the absence of nitrates in the surface layers relevant of Mediterranean oligotrophic spring regime (see Figure 4 lower right panel). These large-scale patterns are in good agreement with observations reported by previous field surveys such as BOUM in 2009 (Moutin and Prieur, 2012) or M84/3 in 2011 (Tanhua et al., 2013). Consequently, the vertical distribution of biomass is marked by a deep chlorophyll maximum; this maximum becomes higher and shallower between the eastern to western basins (see Figure 4 upper right panel). Such spatial contrasts need to be complemented by the temporal evolution of these patterns which can be achieved thanks to the BGC-Argo floats.

[revised manuscript text omitted]

Pasqueron de Fommervault, O., D'Ortenzio, F., Mangin, A., Serra, R., Migon, C., Claustre, H., ... and Schmechtig, C.: Seasonal variability of nutrient concentrations in the Mediterranean Sea: Contribution of Bio-Argo floats, Journal of Geophysical Research: Oceans, 2015.

Ras, J., Claustre, H., and Uitz, J.: Spatial variability of phytoplankton pigment distributions in the Subtropical South Pacific Ocean: Comparison between in situ and predicted data, Biogeosciences, 5(2), 353–369, 2008.

Roesler, C., Uitz, J., Claustre, H., Boss, E., Xing, X., Organelli, E., Briggs, N., Bricaud, A., Schmechtig, C., Poteau, A., D'Ortenzio, F., Ras, J., Drapeau, S., Haëntjens, N. and Barbieux, M.: Recommendations for obtaining unbiased chlorophyll estimates from in-situ chlorophyll fluorometers: A global analysis of WET Labs ECO sensors, Limnology and Oceanography methods, doi: 10.1002/lom3.10185, 2017.

5   Sakamoto, C. M., Johnson, K.S., and Coletti, L.J.: Improved algorithm for the computation of nitrate concentrations in seawater using an in situ ultraviolet spectrophotometer, Limnol Oceanogr-Meth, 7, 132-143, 2009.

Siokou-Frangou, I., Christaki, U., Mazzocchi, M., Montresor, M., D'Alcalà, M.R., Vaqué, D., and Zingone, A.: Plankton in the open Mediterranean Sea: a review, Biogeosciences, 2010.

Schmechtig, C., Poteau, A., Claustre, H., D'Ortenzio, F., and Boss, E.: Processing bio-Argo chlorophyll-a concentration at the

10  DAC level. doi:10.13155/39468, 2015.

Tanhua, T., Hainbucher, D., Schroeder, K., Cardin, V., Álvarez, M., and Civitarese, G.: The Mediterranean Sea system: a review and an introduction to the special issue, Ocean Sci., 9, 789-803, https://doi.org/10.5194/os-9-789-2013, 2013.

Thierry V., Gilbert D., Kobayashi T., Schmid C., and Kanako S.: Processing Argo oxygen data at the DAC level cookbook. http://doi.org/10.13155/39795, 2016.

15  Sea-Bird Scientific: Application Note 64-3: SBE 43 dissolved oxygen (DO) sensor - hysteresis corrections. Available online at : http://www.seabird.com/document/an64-3-sbe-43-dissolved-oxygen-do-sensor-hysteresis-corrections, 2014.

Uchida, H., Kawano, T., Kaneko, I., and Fukasawa, M.: In situ calibration of optode-based oxygen sensors, J. Atmos. Oceanic Technol., 25, 2271–2281, doi:10.1175/2008JTECHO549.1, 2008.

Winkler, L. W.: Die Bestimmung des im Wasser gelosten Sauerstoffes. Ber. Dtsch. Chem. Ges. 21: 2843-2853, 1888.

20  Xing, X., Claustre, H., Blain, S., D'Ortenzio, F., Antoine, D., Ras, J., and Guinet, C.: Quenching correction for in vivo chlorophyll fluorescence acquired by autonomous platforms: A case study with instrumented elephant seals in the Kerguelen region (Southern Ocean), Limnology and Oceanography Methods, 10, 483-495, 2012.

[Figure]

**Figure 1: Cruise track plotted on a time line (colorbar). Port calls marked by red squares, stations are marked by black circles. Zoom of the L-shape track in the eastern coast of Crete.**

| STATION | CAST | DATE UTC | LATITUDE | LONGITUDE | PROFILE DEPTH (m) | BOTTOM DEPTH (m) | # SAMPLE PIGMENTS | #SAMPLE OXYGEN | # SAMPLE NUTRIENTS |
|---|---|---|---|---|---|---|---|---|---|
| LIGURIAN | 1 | 12/5/15 20:10 | 43° 33.52' N | 7° 27.78' E | 1662 | 1684 | 5 | 11 | 11 |
| NORTH IONIAN | 2 | 16/05/15 03:41 | 38° 10.44' N | 18° 30.12' E | 500 | 3038 | 8 | 5 | 11 |
| | 3 | 16/05/15 05:35 | 38° 10.96' N | 18° 30.16' E | 2990 | 3038 | 0 | 11 | 11 |
| CENTRAL LEVANTINE | 4 | 21/05/15 12:21 | 33° 33.90' N | 28° 27.99' E | 500 | 2959* | 8 | 11 | 11 |
| | 5 | 21/05/15 14:14 | 33° 33.76' N | 28° 28.50' E | 1240 | 2959* | 0 | 11 | 11 |
| WEST LEVANTINE | 6 | 22/05/15 10:33 | 34°13.89' N | 24° 49.84' E | 1000 | 2244* | 7 | 11 | 11 |
| | 7 | 22/05/15 15:02 | 34° 12.61' N | 24° 50.76' E | 500 | 2886 | 8 | 11 | 11 |
| | 8 | 22/05/15 17:04 | 34° 12.66' N | 24° 50.56' E | 2871 | 2886 | 0 | 11 | 11 |
| EAST IONIAN | 9 | 26/05/15 12:51 | 36° 41.84' N | 20° 07.32' E | 500 | 3175 | 8 | 11 | 11 |
| | 10 | 26/05/15 14:44 | 36° 41.57' N | 20° 07.21' E | 3165 | 3175 | 0 | 11 | 11 |
| SOUTH TYRRHENIAN | 11 | 30/05/15 10:05 | 39° 10.43' N | 10° 53.47' E | 500 | 2812 | 8 | 11 | 11 |
| | 12 | 30/05/15 13:36 | 39° 11.44' N | 10° 52.37' E | 2803 | 2812 | 0 | 11 | 11 |
| CENTRAL TYRRHENIAN | 13 | 31/05/15 05:21 | 40° 45.22' N | 11° 30.28' E | 500 | 2466 | 8 | 11 | 11 |
| | 14 | 31/05/15 07:14 | 40° 45.87' N | 11° 30.66' E | 2456 | 2466 | 0 | 11 | 11 |

**Table 1: Station summary. For bottom depth, values with asterisk indicate that the measurement has been obtained from the vessel's echo-sounder whether than the altimeter interfaced to the CTD unit.**

| STATION | ARGO WMO | FIRST PROFILE ID | INTER-DISTANCE (km) | INTER-DURATION (h) | TEMP OFFSET (°C) | PSAL OFFEST | OPTODE SLOPE | OPTODE OFFSET ($\mu$mol.kg$^{-1}$) | FLUO N | FLUO R$^2$ | FLUO OFFSET (mg.m$^{-3}$) | FLUO SLOPE | SUNA SLOPE | SUNA OFFSET ($\mu$mol.L$^{-1}$) |
|---|---|---|---|---|---|---|---|---|---|---|---|---|---|---|
| WEST LEV.* | 6901764 | BCN | 0 | 0 | 0.0059 | 0.0150 | 0.9796 | 11.56 | 7 | 0.98 | 0.01 | 0.67 | 1.00 | 3.20 |
| WEST LEV. | 6901765 | 000 | 1 | 19 | 0.0003 | 0.0031 | 1.0660 | 3.26 | 8 | 0.77 | 0.04 | 0.62 | 1.00 | 4.00 |
| WEST LEV.* | 6901766 | BCN | 0 | 0 | 0.0053 | 0.0081 | 1.0275 | 6.30 | 7 | 0.98 | -0.02 | 0.65 | 1.11 | 0.80 |
| CENTRAL TYR. | 6901767 | 000 | 3 | 7 | | | | | 8 | 0.86 | 0.03 | 0.49 | 1.00 | -2.80 |
| CENTRAL TYR. | 6901767 | 001 | 3 | 29 | 0.0021 | -0.0009 | 1.1045 | -3.59 | | | | | 1.00 | -2.70 |
| NORTH ION. | 6901768 | 001 | 12 | 31 | 0.0052 | 0.0009 | 1.0235 | 6.66 | 8 | 0.89 | 0.04 | 0.63 | 1.00 | 2.10 |
| SOUTH TYR. | 6901769 | 000 | 2 | 26 | 0.0214 | 0.0050 | 1.1626 | -14.87 | 8 | 0.82 | 0.03 | 0.58 | 1.00 | 3.90 |
| EAST ION. | 6901771 | 000 | 2 | 21 | 0.0085 | 0.0042 | 1.0658 | 0.51 | 8 | 0.93 | 0.02 | 0.55 | 1.17 | 0.10 |
| CENTRAL LEV. | 6901773 | 000 | 3 | 22 | 0.0067 | 0.0070 | 1.0923 | -2.40 | 8 | 0.99 | 0.01 | 0.51 | | |
| AVERAGE | | | 3 | 17 | 0.0069 | 0.0053 | 1.0652 | 0.93 | | | 0.02 | 0.59 | 1.04 | 1.08 |
| STD | | | 4 | 12 | 0.0064 | 0.0049 | 0.0564 | 8.12 | | | 0.02 | 0.07 | 0.07 | 2.74 |

* *metrological verification exercise: deployed at another location than the station*

**Table 2: BGC-Argo float summary. For every BGC-Argo float deployed with a CTD cast of reference, the distance and duration with the first profile of the float is indicated. The results of metrological verification by parameter is reported. STD stands for standard deviation.**

| Cast | Depth (m) | BT distance (m) | Error velocity (cm.s$^{-1}$) | | | LDEO final parameters | Misfits L ms S (cm.s$^{-1}$) | Comments |
|---|---|---|---|---|---|---|---|---|
| | | | L without S constraint | L with S constraint | S | | | |
| 1 | 1721 | 26 | 2.5 | 2.5 | 5.5 | L+S+BT | 1.8 | |
| 2 | 498 | | 3.4 | 3.4 | 5.7 | L+S | 3.8 | |
| 3 | 2990 | 53 | 20.3 | 18.9 | 6.5 | L+S+BT | 19.2 | rough sea, high tilt |
| 4 | 501 | | 3.1 | 2.3 | 6.9 | L+S | 3.1 | |
| 5 | 1243 | | 2.9 | 3.4 | 6.4 | L+S | 6.9 | |
| 6 | 996 | | 2.5 | 2.6 | 5 | L+S | 2.0 | |
| 7 | 496 | | 2.5 | 2.4 | 4.6 | L+S | 2.2 | |
| 8 | 2871 | 16 | 2.9 | 4.8 | 4.7 | L+S+BT | 6.1 | |
| 9 | 502 | | 2.2 | 3.1 | 4.5 | downlooker+S | 2.6 | uplooker failed, low battery |
| 10 | 3165 | 17 | 20.7 | 50.4 | 5.2 | downlooker+S+BT | 36.0 | |
| 11 | 497 | | 3 | 3 | 5.9 | L+S | 4.5 | |
| 12 | 2805 | 7 | 5.4 | 4.2 | 6.1 | L+S+BT | 4.4 | |
| 13 | 505 | | 2.6 | 2.5 | 5 | L+S | 2.5 | |
| 14 | 2456 | 12 | 2.8 | 2.8 | 5.3 | L+S+BT | 3.6 | |

**Table 3. Summary of ocean current profiles collected at the stations. Depth and bottom track (BT) distance, when available, are indicated. Error velocities were computed for three sets of profiles: LADCP (L) data only, SADCP (S) data only, L data processed under the constraint of S data. Final process parameters were chosen in function that lead to the misfits between L (with final process parameters) and S currents.**

[Figure]

5    **Figure 2: Oxygen residuals between sensor and Winkler measurements, plotted in function of time (upper panels) and in function of depth (lower panels). The residuals for the electrochemical sensor are plotted in the left panels, the ones for the optode in the right panels.**

| Pigment | Variable name | Units | detection wavelength (nm) | Limit of Detection ng/inj | Limit of Detection for 2L filtered (in mg.m$^{-3}$) |
|---|---|---|---|---|---|
| Chlorophyll c3 | CHLC3 | mg.m$^{-3}$ | 450 | 0.015 | 0.0002 |
| Chlorophyll c1+c2 | CHLC2 | mg.m$^{-3}$ | 450 | 0.018 | 0.0002 |
| Sum Chlorophyllide a | CHLDA | mg.m$^{-3}$ | 667 | 0.016 | 0.0002 |
| Peridinin | PERI | mg.m$^{-3}$ | 450 | 0.007 | 0.0001 |
| Sum Phaeophorbid a | PHDA | mg.m$^{-3}$ | 667 | 0.009 | 0.0001 |
| 19'-Butanoyloxyfucoxanthin | BUT | mg.m$^{-3}$ | 450 | 0.009 | 0.0001 |
| Fucoxanthin | FUCO | mg.m$^{-3}$ | 450 | 0.009 | 0.0001 |
| Neoxanthin | NEO | mg.m$^{-3}$ | 450 | 0.009 | 0.0001 |
| Prasinoxanthin | PRAS | mg.m$^{-3}$ | 450 | 0.009 | 0.0001 |
| Violaxanthin | VIOLA | mg.m$^{-3}$ | 450 | 0.012 | 0.0001 |
| 19'-Hexanoyloxyfucoxanthin | HEX | mg.m$^{-3}$ | 450 | 0.009 | 0.0001 |
| Diadinoxanthin | DIADINO | mg.m$^{-3}$ | 450 | 0.014 | 0.0002 |
| Alloxanthin | ALLO | mg.m$^{-3}$ | 450 | 0.015 | 0.0002 |
| Diatoxanthin | DIATO | mg.m$^{-3}$ | 450 | 0.015 | 0.0002 |
| Zeaxanthin | ZEA | mg.m$^{-3}$ | 450 | 0.014 | 0.0002 |
| Lutein | LUT | mg.m$^{-3}$ | 450 | 0.014 | 0.0002 |
| Bacteriochlorophyll a | BCHLA | mg.m$^{-3}$ | 770 | 0.010 | 0.0001 |
| Divinyl Chlorophyll b | DVCHLB | mg.m$^{-3}$ | 450 | 0.004 | 0.0001 |
| Chlorophyll b | CHLB | mg.m$^{-3}$ | 450 | 0.004 | 0.0001 |
| Total Chlorophyll b | TCHLB | mg.m$^{-3}$ | 450 | 0.004 | 0.0001 |
| Divinyl Chlorophyll a | DVCHLA | mg.m$^{-3}$ | 667 | 0.011 | 0.0001 |
| Chlorophyll-a | CHLA | mg.m$^{-3}$ | 667 | 0.011 | 0.0001 |
| Total Chlorophyll a | TCHLA | mg.m$^{-3}$ | 667 | 0.011 | 0.0001 |
| sum Phaeophytin a | PHYTNA | mg.m$^{-3}$ | 667 | 0.007 | 0.0001 |
| Sum Carotenes | TCAR | mg.m$^{-3}$ | 450 | 0.013 | 0.0002 |

**Table 4: list of parameters in the pigment dataset, the name of variable, the units, and for each pigment, the detection wavelengths and the associated limits of detection in ng per injection.**

[Figure]

**Figure 3: Velocity distribution of the upper water column along a west–east section through the Mediterranean Sea. Data are recorded by SADCP. Inner panel indicates the location of the ship track and the section. Grey areas: no data are available.**

[Figure]

**Figure 4: TS diagram determined by CTD data (upper left); total chlorophyll-a concentration profiles by HPLC method (upper right); Dissolved oxygen concentration profiles by CTD data (lower left); Nitrate concentration profiles by colorimetric method (lower right). The inner panel shows the location of CTD stations.**